# ✂ Mixing Importance with Diversity: Joint Optimization for KV Cache Compression in Large Vision-Language Models

**Xuyang Liu**[1,2*]   **Xiyan Gui**[1,3*]   **Yuchao Zhang**[1]   **Linfeng Zhang**[1✉]

[1]EPIC Lab, Shanghai Jiao Tong University    [2]Sichuan University
[3]Huazhong University of Science and Technology
**Homepage:** **https://xuyang-liu16.github.io/MixKV/**

## Abstract

Recent large vision-language models (LVLMs) demonstrate remarkable capabilities in processing extended multi-modal sequences, yet the resulting key-value (KV) cache expansion creates a critical memory bottleneck that fundamentally limits deployment scalability. While existing KV cache compression methods focus on retaining high-importance KV pairs to minimize storage, they often overlook the modality-specific semantic redundancy patterns that emerge distinctively in multi-modal KV caches. In this work, we first analyze how, beyond simple importance, the KV cache in LVLMs exhibits varying levels of redundancy across attention heads. We show that relying solely on importance can only cover a subset of the full KV cache information distribution, leading to potential loss of semantic coverage. To address this, we propose `MixKV`, a novel method that mixes importance with diversity for optimized KV cache compression in LVLMs. `MixKV` adapts to head-wise semantic redundancy, selectively balancing diversity and importance when compressing KV pairs. Extensive experiments demonstrate that `MixKV` consistently enhances existing methods across multiple LVLMs. Under extreme compression (budget=64), `MixKV` improves baseline methods by an average of **5.1%** across five multi-modal understanding benchmarks and achieves remarkable gains of **8.0%** and **9.0%** for SnapKV and AdaKV on GUI grounding tasks, all while maintaining comparable inference efficiency. Furthermore, `MixKV` extends seamlessly to LLMs with comparable performance gains.

## 1 Introduction

Large vision-language models (LVLMs) (Li et al., 2024a; Chen et al., 2024d) have achieved remarkable performance in multimodal understanding by effectively integrating visual information and user instructions into the input space of large language models (LLMs) (Grattafiori et al., 2024; Yang et al., 2025a). With the growing demand for understanding long-context visual inputs, including high-resolution images (Bai et al., 2025b; Zhu et al., 2025) and long videos (Shu et al., 2025; Qin et al., 2025), LVLMs must process an increasing number of visual tokens. However, processing such long-context inputs generates numerous key-value (KV) pairs in the KV cache of LLMs, substantially increasing GPU memory consumption and degrading computational efficiency due to memory access latency and bandwidth constraints (Liu et al., 2025c; Shao et al., 2025b).

To address the KV storage overhead, two main approaches have emerged. Token compression methods (Yang et al., 2025b; Chen et al., 2024a; Liu et al., 2025b) directly compress visual tokens to indirectly reduce KV cache storage, but often underperform in high-resolution fine-grained tasks like text understanding (Singh et al., 2019) and document processing (Mathew et al., 2021). More effective KV cache compression methods directly evict KV pairs in the LLM to minimize storage while preserving performance (Wan et al., 2024; Li et al., 2024b), thereby enhancing decoding efficiency and throughput. However, current KV cache compression methods for LVLMs predominantly rely on attention-based importance scores to decide which KV pairs to retain (Tao et al., 2025a; Wang et al., 2025b). While this strategy does reduce the KV cache size, it fails to consider the intrinsic semantic characteristics of KV pairs in multi-modal settings. To bridge this gap, we conduct a comprehensive analysis and identify **two key characteristics** of KV pairs in LVLMs:

---

*Equal contribution. ✉Corresponding author: zhanglinfeng@sjtu.edu.cn.

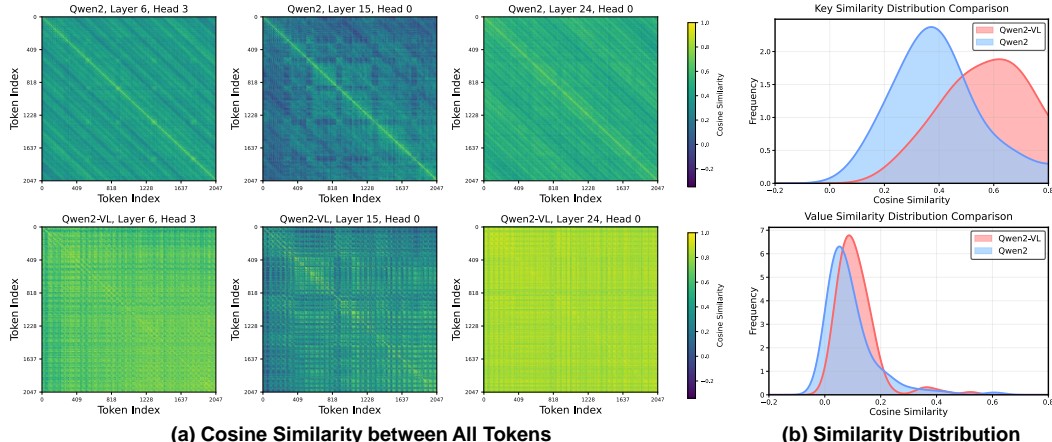

Figure 1: **Visualization of KV cache redundancy across different models.** **(a)** presents the similarity of keys across the same layer and head, with Qwen2-VL (bottom) processing vision-language information and Qwen2 (top) handling pure text information. **(b)** quantifies the average similarity distribution of keys (top) and values (bottom) across all layers and heads for Qwen2 and Qwen2-VL.

**(I) Vision-Language Redundancy Differences:** Visual information in LVLMs contains significantly more semantic redundancy than textual information in LLMs. Images often contain repetitive visual elements (*e.g.*, similar textures, repeated patterns), leading to higher semantic similarity among KV pairs during vision-language processing. Figure 1 provides compelling evidence: **(a)** shows that Qwen2-VL exhibits much denser high-similarity regions compared to the more diverse patterns of Qwen2, while **(b)** reveals that keys in Qwen2 peak around 0.2-0.4 average similarity whereas Qwen2-VL keys peak around 0.6-0.8, a **2-3× increase**. This demonstrates that *KV pairs in LVLMs exhibit substantially higher semantic redundancy than in LLMs*.

**(II) Head-wise Redundancy Differences:** Within LVLMs, different attention heads focus on distinct multi-modal aspects (Wang et al., 2025b). Some heads capture global features with lower redundancy, while others focus on local details with higher semantic similarity. Figure 2 illustrates this phenomenon across multiple tasks: for Qwen2-VL-7B, certain heads show extremely high average similarity exceeding **0.9**, while other heads maintain relatively low similarity below **0.3**. This pattern is consistent across different vision-language tasks, indicating that *KV pairs in LVLMs show varying degrees of semantic redundancy across attention heads in the LLM*.

Furthermore, our analysis reveals that importance-based compression methods fail to fully replicate the information distribution of the original KV cache, leading to potential information loss (Figure 3). Therefore, we argue that beyond importance, preserving diverse KV pairs at per-head granularity is essential for minimizing semantic redundancy while maintaining comprehensive information coverage. To this end, we propose `MixKV`, which adopts a principled ***"mixing importance with diversity"*** approach. Specifically, `MixKV` extends existing importance-based KV compression methods by incorporating head-wise semantic diversity evaluation. By independently measuring semantic similarity within each attention head, `MixKV` adaptively balances importance and diversity per head to achieve fine-grained joint optimization of KV cache compression in LVLMs.

`MixKV` is a plug-and-play framework that enhances existing KV compression methods with consistent performance gains, maintaining inference efficiency while better preserving the distributional properties of the original KV cache. In summary, the **main contributions** are as follows:

1. **Semantic Redundancy Analysis.** We conduct in-depth analyzes of KV caches in LVLMs, revealing substantial inherent semantic redundancy. Besides, we demonstrate that importance-based methods fail to preserve full KV distribution coverage, exposing fundamental limitations.
2. **Mixing Importance with Diversity.** Based on our analysis, we propose `MixKV`, a head-wise adaptive mechanism that quantifies semantic redundancy to create principled weighting between importance and diversity scores for the joint optimization of KV cache compression.
3. **Comprehensive Experimental Validation.** Extensive experiments across diverse multi-modal and text benchmarks demonstrate that `MixKV` yields consistent performance improvements for existing importance-based compression methods while maintaining inference efficiency.

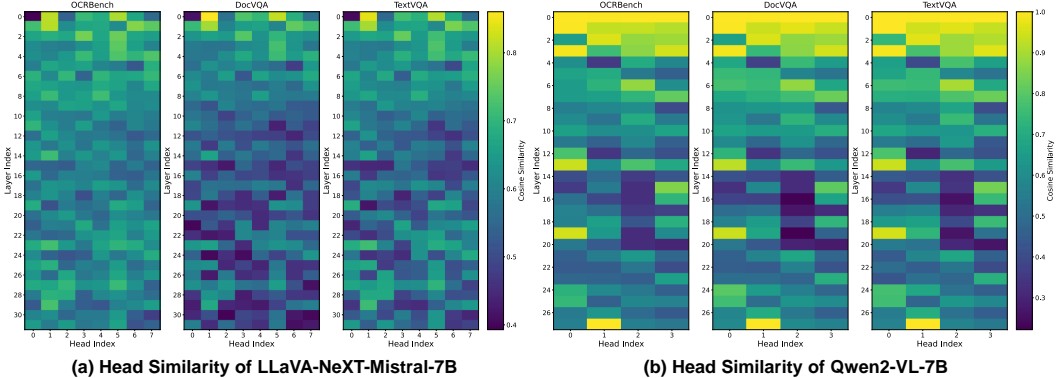

Figure 2: **Visualization of KV cache redundancy across different heads in LLMs.** **(a)** and **(b)** present the average cosine similarity across heads of LLaVA-NeXT-Mistral-7B and Qwen2-VL-7B within the LLMs, with brighter heads indicating greater similarity in semantic information.

## 2 RELATED WORK

**Large Vision-language Models.** Current large vision-language models (LVLMs) integrate a vision encoder (*i.e.*, ViT), a projector module, and a large language model (LLM) to enable multi-modal comprehension (Liu et al., 2023; 2024a). To meet the growing demand for high-resolution image understanding, recent LVLMs introduce higher-resolution inputs via dynamic cropping strategies, such as LLaVA-NeXT (Liu et al., 2024b) and InternVL series (Chen et al., 2024c;b), or native resolution processing like Qwen2-VL (Wang et al., 2024; Bai et al., 2025b) and GLM-4.5V (Hong et al., 2025). Additionally, video large language models (VideoLLMs) such as LLaVA-Video (Zhang et al., 2024) and Video-XL-2 (Qin et al., 2025) process multi-frame videos with thousands of frames. This trend dramatically increases visual token counts, leading to substantial computational costs and GPU memory burdens due to the key-value (KV) cache in the attention mechanism.

**Long-Context Optimization.** Longer contexts generally improve the performance of LVLMs and enable more comprehensive multi-modal understanding (Wang et al., 2025c; Chen et al., 2025). Extensive work aims to make long-context processing more efficient and can be broadly categorized into three directions: **(i)** *Efficient computational architectures*, such as sparse attention (Li et al., 2025b; Xu et al., 2025), linear attention (Li et al., 2025a), and state-space models (Gu & Dao, 2024), which reduce the quadratic complexity of attention with respect to sequence length and thus accelerate long-context processing; **(ii)** *Model-centric compression*, including network pruning (Ma et al., 2023), model quantization (Wang et al., 2025a), and knowledge distillation (Cai et al., 2025a), which reduce parameter count and thereby lower the computational and memory cost of long-context inference; and **(iii)** *Data-centric compression*, which reduces the effective context length or storage processed by the model, for example via token compression (Yang et al., 2025b; Shao et al., 2025a), KV cache compression (Li et al., 2024b; Wang et al., 2025b) or KV cache quantization (Liu et al., 2024d; Tao et al., 2025b), thus directly improving the efficiency of long-context computation. These three directions optimize long-context processing from complementary perspectives and are largely orthogonal to each other. Given that the context lengths required by modern applications have rapidly increased (Liu et al., 2025c), in this work we focus on the data-centric perspective and compress the stored KV cache to enable efficient long-context computation for LVLMs.

**KV Cache Compression.** The KV cache stores computed key-value (KV) pairs during the LLM's pre-filling phase to avoid redundant computations in decoding and enhance inference efficiency. However, long-context multi-modal inputs impose a significant GPU memory burden on the KV cache. To alleviate this, several works propose KV cache compression techniques, categorized as: **(i)** *vision token compression* that directly compresses vision tokens (Chen et al., 2024a; Liu et al., 2025a), and **(ii)** *KV cache compression* that compresses stored KV pairs during pre-filling (Zhang et al., 2023; Liu et al., 2024e). Current KV compression methods are mainly designed for LLMs, such as SnapKV (Li et al., 2024b), which clusters important KV positions using attention patterns from an observation window; KNorm (Devoto et al., 2024), which uses $\ell_2$ key norms to score and retain KV pairs with lower norms; and AdaKV (Feng et al., 2025), which adaptively allocates eviction budgets across attention heads. Methods specifically designed for LVLMs include InfiniPot-V (Kim et al., 2025), which employs Value Norm for KV pair selection, and SparseMM (Wang et al.,

2025b), which allocates asymmetric budgets across attention heads based on their importance and retains high-attention KV pairs from observation windows. Most existing compression methods follow the paradigm of retaining critical KV pairs and evicting less important ones.

Unlike prior works that focus primarily on importance-based selection, we identify a critical characteristic in LVLMs: ***heterogeneous head-wise redundancy***, where KV caches exhibit varying degrees of semantic redundancy across attention heads (Figure 2), causing importance-only methods to retain KV pairs that fail to cover the full information spectrum in high-redundancy heads. This motivates us to jointly consider importance and diversity for more effective KV cache compression.

## 3 METHODOLOGY

### 3.1 PRELIMINARY: LARGE VISION-LANGUAGE MODELS

**LVLM Architecture.** Contemporary large vision-language models (LVLMs) generally adopt a "ViT-Projector-LLM" architecture (Li et al., 2024a; Wang et al., 2024), which consists of three primary components. For an input image $\mathbf{I} \in \mathbb{R}^{H \times W \times 3}$ or video $\mathbf{V} = \{\mathbf{v}_i\}_{i=1}^T \in \mathbb{R}^{T \times H \times W \times 3}$: **(i)** The visual encoder (*i.e.*, ViT) extracts visual features, yielding embeddings $\mathbf{E} \in \mathbb{R}^{N \times D}$ for images or $\mathbf{E} = \{\mathbf{e}_i\}_{i=1}^T \in \mathbb{R}^{T \times N \times D}$ for videos; **(ii)** A projection layer, often a two-layer MLP, maps these to vision tokens $\mathbf{F}^v \in \mathbb{R}^{M \times D'}$ for images or $\mathbf{F}^v = \{\mathbf{f}_i^v\}_{i=1}^T \in \mathbb{R}^{T \times M \times D'}$ for videos, with $M \leq N$; and **(iii)** The LLM processes the combined visual and text tokens $\mathbf{F}^t$ in two phases: During the *pre-filling phase*, it computes KV pairs for all input tokens ($\mathbf{F}^v$ and $\mathbf{F}^t$) and stores them in the KV cache to avoid redundant computations; in the *decoding phase*, it auto-regressively generates responses, leveraging the KV cache for efficient retrieval of prior KV pairs in attention mechanisms:

$$p\left(\mathbf{Y} \mid \mathbf{F}^v, \mathbf{F}^t\right) = \prod_{j=1}^L p\left(\mathbf{y}_j \mid \mathbf{F}^v, \mathbf{F}^t, \mathbf{Y}_{1:j-1}; \mathcal{C}\right), \tag{1}$$

where $\mathbf{Y} = \{\mathbf{y}_j\}_{j=1}^L$ is the output sequence, and $\mathcal{C}$ denotes the KV cache. Thus, LVLMs achieve multi-modal understanding based on visual inputs and user instructions.

**KV Cache Compression.** Multi-modal long-context sequences result in numerous KV pairs, which lead to significant memory overhead. KV cache compression addresses this challenge by introducing a compression operator $\mathbf{\Phi}$, which selectively reduces the number of stored KV cache (Wang et al., 2025b). Typically, $\mathbf{\Phi}$ involves an evaluation function $\mathcal{E}$ that assigns scores $s_i = \mathcal{E}(\mathbf{K}_{h,i}^l, \mathbf{V}_{h,i}^l)$ to each KV pair $i$, with compression based on these scores by retaining top-$b$ highest-scoring pairs given a budget $B$, such that for each layer $l$ and head $h$, KV pairs $\mathbf{K}_h^l, \mathbf{V}_h^l \in \mathbb{R}^{T \times D}$ ($T$ is sequence length, $D$ dimension per head) are compressed into compact representations $\hat{\mathbf{K}}_h^l, \hat{\mathbf{V}}_h^l = \text{TopB}(\mathbf{K}_h^l, \mathbf{V}_h^l, \{s_i\}_{i=1}^T)$. KV cache compression targets KV tensors computed during pre-filling, reducing memory burden while supporting efficient attention in decoding.

### 3.2 ANALYSIS OF KV PAIRS CHARACTERISTICS

To optimize KV cache compression in LVLMs, we begin by analyzing key characteristics of the KV cache. An intuitive characteristic is importance, aimed at retaining KV pairs with greater significance while compressing those with lesser importance, thereby enabling efficient compression.

**Importance Metrics.** Existing methods evaluate KV pair importance from two perspectives, *intrinsic* and *extrinsic*, each employing distinct metrics to compute importance scores $s_{\text{imp}}$:

- **Intrinsic Importance:** Determined by inherent KV vector properties, including Key Norm (KNorm) (Devoto et al., 2024), which computes the $\ell_2$ norm of each key vector with its negative assigned as $s_{\text{imp},i}^{\text{in}} = -s_{\text{imp},i}^{\text{in (KNorm)}}$ for compression scoring, and Value Norm (VNorm) (Kim et al., 2025), which calculates the $\ell_2$ norm of each value vector, using it directly as $s_{\text{imp},i}^{\text{in}} = s_{\text{imp},i}^{\text{in (VNorm)}}$ for scoring. Compression retains the KV pairs with higher $s_{\text{imp},i}^{\text{in}}$, such that $\hat{\mathbf{K}}_h^l, \hat{\mathbf{V}}_h^l = \text{TopB}(\mathbf{K}_h^l, \mathbf{V}_h^l, \{s_{\text{imp},i}^{\text{in}}\}_{i=1}^T)$, where $s_{\text{imp},i}^{\text{in}}$ represents the intrinsic importance.
- **Extrinsic Importance:** Quantified by average attention scores from an observation window at the prompt end (default length 32) (Li et al., 2024b; Cai et al., 2025b), reflecting instruction

relevance. The score $s_{\text{imp},i}^{\text{ex}} = \frac{1}{|\text{window}|} \sum_{j \in \text{window}} \text{Attention}(\mathbf{Q}_j, \mathbf{K}_i)$ is computed from attention weights between query $\mathbf{Q}_j$ and key $\mathbf{K}_i$, balancing modalities effectively. This score serves as $s_{\text{imp},i}^{\text{ex}}$ for compression. Compression prioritizes pairs with higher $s_{\text{imp},i}^{\text{ex}}$, such that $\hat{\mathbf{K}}_h^l, \hat{\mathbf{V}}_h^l = \text{TopB}(\mathbf{K}_h^l, \mathbf{V}_h^l, \{s_{\text{imp},i}^{\text{ex}}\}_{i=1}^T)$, where $s_{\text{imp},i}^{\text{ex}}$ primarily reflects instruction relevance.

We argue that a comprehensive assessment of KV pair importance requires **integrating both intrinsic and extrinsic perspectives**. This ensures a balanced evaluation of inherent significance and instruction relevance. Specifically, the integrated importance score is computed as:

$$s_{\text{imp},i} = s_{\text{imp},i}^{\text{ex}} + s_{\text{imp},i}^{\text{in}} \qquad (2)$$

where we employ $s_{\text{imp},i}^{\text{in (VNorm)}}$ as the default intrinsic component. To ensure compatibility between VNorm and attention-based extrinsic importance, we normalize VNorm scores to $[0, 1]$ and scale them to match attention score magnitudes: $s_{\text{scaled},i}^{\text{in}} = s_{\text{norm},i}^{\text{in}} \cdot \frac{\bar{s}_{\text{imp}}^{\text{ex}}}{\bar{s}_{\text{norm}}^{\text{in}} + \epsilon}$, where $\bar{s}_{\text{imp}}^{\text{ex}}$ and $\bar{s}_{\text{norm}}^{\text{in}}$ denote respective mean values. Comprehensive ablation studies in Section 4.3 Table 4 validate the effectiveness of this design choice. However, methods relying solely on importance suffer from a critical limitation: *they preferentially retain semantically similar KV pairs, leading to significant redundancy in the compressed cache and loss of global semantic coverage*.

Figure 3 further presents this limitation, which performs a t-SNE visualization of KV cache distributions. We observe that methods relying only on importance, such as SnapKV (Li et al., 2024b) (blue stars), *fail to adequately cover the full KV cache information*. In Figure 3, SnapKV primarily focuses on a small portion of the information, losing semantic coverage compared to the full KV distribution (light gray circles). This is because importance-based methods prioritize task-relevant, highly similar information, often neglecting the broader diversity of KV pairs. As a result, relying solely on importance introduces redundancy by retaining semantically similar KV pairs, which do not provide the full semantic richness of the KV cache. Therefore, effective KV cache compression in LVLMs requires incorporating diversity to retain non-redundant KV pairs. This enables KV cache compression methods to **approximate the full original semantic distribution** of the uncompressed KV cache more effectively.

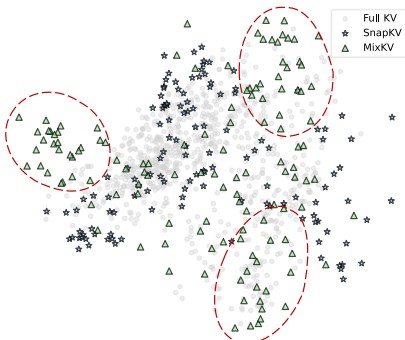

Figure 3: **t-SNE visualization of KV cache distributions under different settings.** "Full KV" represents the original KV distribution of Qwen2-VL without compression, serving as the reference distribution.

**Diversity Metrics.** Beyond importance, semantic diversity serves as another crucial characteristic for effective KV cache compression in LVLMs. We focus on key diversity, as keys primarily govern the attention patterns and semantic focus within a head, making them the most direct levers for controlling information redundancy. To quantify diversity in a computationally efficient manner, we adopt the *negative cosine similarity* between each key and the global average key, as it serves as an intuitive proxy for capturing the breadth of the semantic distribution. For each layer $l$ and head $h$, we first normalize each key vector: $\hat{\mathbf{K}}_{h,i}^l = \frac{\mathbf{K}_{h,i}^l}{\|\mathbf{K}_{h,i}^l\|}$, where $\mathbf{K}_{h,i}^l \in \mathbb{R}^D$ represents the $i$-th key vector and $T$ is the sequence length. We then compute a single global key representation via averaging: $\bar{\hat{\mathbf{K}}}_h^l = \frac{1}{T} \sum_{i=1}^T \hat{\mathbf{K}}_{h,i}^l$. This ensures that the diversity scores, computed via cosine similarity, are obtained in *linear time* with respect to $T$. Specifically, the diversity score for each KV pair is $s_i^{\text{div}} = -\hat{\mathbf{K}}_{h,i}^l \cdot \bar{\hat{\mathbf{K}}}_h^l$. Here, higher scores indicate greater distinctiveness from the global pattern, as the negative cosine similarity encourages the retention of diverse information.

## 3.3 HEAD-WISE ADAPTIVE MIXING MECHANISM

Section 3.2 analyzes importance metrics $s_{\text{imp}}$ and diversity metrics $s_{\text{div}}$, both critical for compression. A natural question arises: ***How can these complementary metrics be combined effectively?*** While simply adding the importance and diversity scores ($s_{\text{imp}} + s_{\text{div}}$) offers simplicity, Figure 2 shows that varying semantic similarity levels across attention heads make uniform mixing sub-optimal.

Our central insight is that heads exhibiting higher semantic redundancy should prioritize diversity preservation to prevent similar KV pair retention, while less redundant heads can emphasize importance-based selection. This motivates the head-wise adaptive weighting mechanism in `MixKV`.

**Head-wise Redundancy Quantification.** We develop a principled approach to quantify semantic redundancy within each attention head. For layer $l$ and head $h$, we employ the off-diagonal average similarity of normalized key vectors as our redundancy measure.

Using the normalized key matrix $\hat{\mathbf{K}}_h^l \in \mathbb{R}^{T \times D}$ from diversity computation, we construct the similarity matrix $\mathbf{R}_h^l = \hat{\mathbf{K}}_h^l (\hat{\mathbf{K}}_h^l)^T \in \mathbb{R}^{T \times T}$. By exploiting the relationship between dot products and norms, the total similarity sum becomes:

$$\sum_{i,j=1}^{T} \mathbf{R}_{h,i,j}^l = \left( \sum_{i=1}^{T} \hat{\mathbf{K}}_{h,i}^l \right) \cdot \left( \sum_{j=1}^{T} \hat{\mathbf{K}}_{h,j}^l \right) = T^2 \|\bar{\hat{\mathbf{K}}}_h^l\|_2^2 \qquad (3)$$

where $\bar{\hat{\mathbf{K}}}_h^l = \frac{1}{T} \sum_{i=1}^{T} \hat{\mathbf{K}}_{h,i}^l$. Given that normalized vectors yield unit diagonal elements due to the normalization process, the off-diagonal average similarity is:

$$\bar{r}_h^l = \frac{T^2 |\bar{\hat{\mathbf{K}}}_h^l|_2^2 - T}{T(T-1)}. \qquad (4)$$

This formulation ensures that as $\bar{r}_h^l \to 1$ (high redundancy), diversity weight increases to prevent redundant retention, while $\bar{r}_h^l \to 0$ (low redundancy) prioritizes importance-based selection. This mathematical property positions $\bar{r}_h^l$ as an ideal adaptive weight for diversity scores.

**Head-wise Adaptive Mixing.** Based on the redundancy quantification, we develop the *head-wise adaptive mixing function* $W^{\text{head}}(\cdot)$. To ensure scale compatibility between importance and diversity scores, we first normalize diversity scores to $[0, 1]$: $\tilde{s}_i^{\text{div}} = \frac{s_i^{\text{div}} - \min_j(s_j^{\text{div}})}{\max_j(s_j^{\text{div}}) - \min_j(s_j^{\text{div}}) + \epsilon}$, then scale them to match the magnitude of importance scores: $s_{\text{scaled},i}^{\text{div}} = \tilde{s}_i^{\text{div}} \cdot \frac{\bar{s}_{\text{imp}}}{\bar{s}_{\text{div}} + \epsilon}$.

The comprehensive score is computed through our *head-wise adaptive mixing function* $W^{\text{head}}(\cdot)$:

$$s_i^{\text{comp}} = W^{\text{head}}(s_{\text{imp}} + s_{\text{div}}) = (1 - \bar{r}_h^l) \cdot s_{\text{imp},i} + \bar{r}_h^l \cdot s_{\text{scaled},i}^{\text{div}} \qquad (5)$$

Through this formulation, `MixKV` achieves adaptive adjustment: redundant heads ($\bar{r}_h^l \to 1$) emphasize diverse KV pairs, while less redundant heads ($\bar{r}_h^l \to 0$) prioritize importance. This head-wise adaptation ensures the compressed KV cache preserves both critical information and semantic diversity. KV compression is realized by selecting the top-$B$ pairs with the highest comprehensive scores: $\hat{\mathbf{K}}_h^l, \hat{\mathbf{V}}_h^l = \text{TopB}(\mathbf{K}_h^l, \mathbf{V}_h^l, \{s_i^{\text{comp}}\}_{i=1}^{T})$. Notably, Figure 3 demonstrates that the adaptive mixing strategy of `MixKV` enables SnapKV to leverage KV cache diversity, thereby capturing a broader range of information and encompassing a wider distribution of the full KV cache (highlighted in red circles). See Figure 7 for more visualizations and Appendix A.6 for the algorithm.

## 4 EXPERIMENTS

### 4.1 EXPERIMENTAL SETTING

**Model Details.** We evaluate `MixKV` across a diverse set of architectures to ensure generalizability: LLaVA-NeXT-Mistral-7B (Liu et al., 2024b), InternVL3-8B (Li et al., 2024a), and Qwen2-VL-7B-Instruct (Wang et al., 2024) for multi-modal understanding tasks; Qwen2.5-VL-7B-Instruct (Bai et al., 2025b) for GUI grounding tasks; and Mistral-7B-Instruct-v0.2 (Jiang et al., 2023) and Llama-3.1-8B-Instruct (Grattafiori et al., 2024) for text-only evaluation.

**Benchmark Details.** We select a range of multi-modal understanding benchmarks and a comprehensive text understanding benchmark for evaluation. For image understanding, we include DocVQA (Mathew et al., 2021), OCRBench (Liu et al., 2024c), TextVQA (Singh et al., 2019), ChartQA (Masry et al., 2022), and TextCaps (Sidorov et al., 2020), along with ScreenSpot-v2 (Wu et al., 2024) for GUI grounding. For text understanding, we adopt LongBench (Bai et al., 2024).

Table 1: **Performance on multiple image understanding benchmarks.** Since SparseMM (Wang et al., 2025b) does not provide head importance scores for InternVL3-8B (Zhu et al., 2025), we cannot reproduce their results on this model. "Full KV" means caching all KV pairs (upper bound).

| Methods | DocVQA (%) | | | OCRBench (%) | | | TextVQA (%) | | | ChartQA (%) | | | TextCaps | | |
|---|---|---|---|---|---|---|---|---|---|---|---|---|---|---|---|
| | 256 | 128 | 64 | 256 | 128 | 64 | 256 | 128 | 64 | 256 | 128 | 64 | 256 | 128 | 64 |
| *LLaVA-NeXT-Mistral-7B* | | | | | | | | | | | | | | | |
| *Full KV* | | 63.6 | | | 52.9 | | | 65.7 | | | 52.9 | | | 0.707 | |
| **SnapKV** | 59.7 | 55.2 | 47.3 | 45.0 | 39.0 | 31.9 | 63.5 | 61.0 | 57.1 | 50.2 | 47.5 | 42.7 | 0.650 | 0.558 | 0.444 |
| + MixKV | **61.7** | **58.1** | **48.8** | **49.9** | **44.7** | **36.1** | **65.2** | **64.3** | **60.1** | **50.8** | **47.7** | **43.6** | **0.708** | **0.659** | **0.514** |
| $\Delta_{baseline}$ | **+2.0** | **+2.9** | **+1.5** | **+4.9** | **+5.7** | **+4.2** | **+1.7** | **+3.3** | **+3.0** | **+0.6** | **+0.2** | **+0.9** | **+0.058** | **+0.101** | **+0.070** |
| **PyramidKV** | 58.2 | 54.3 | 43.4 | 44.1 | 39.4 | 29.1 | 62.9 | 60.9 | 54.8 | 49.1 | 47.1 | 40.8 | 0.621 | 0.553 | 0.407 |
| + MixKV | **60.8** | **57.2** | **45.1** | **49.7** | **43.7** | **32.0** | **64.9** | **63.8** | **57.8** | **50.8** | **47.5** | **41.3** | **0.687** | **0.656** | **0.466** |
| $\Delta_{baseline}$ | **+2.6** | **+2.9** | **+1.7** | **+5.6** | **+4.3** | **+2.9** | **+2.0** | **+2.9** | **+3.0** | **+1.7** | **+0.4** | **+0.5** | **+0.066** | **+0.103** | **+0.059** |
| **AdaKV** | 59.6 | 55.9 | 48.7 | 45.1 | 40.4 | 32.8 | 62.9 | 60.5 | 56.9 | 50.4 | 47.8 | 44.6 | 0.646 | 0.566 | 0.440 |
| + MixKV | **61.3** | **58.3** | **50.8** | **49.8** | **44.9** | **36.6** | **65.3** | **63.7** | **59.6** | **50.9** | **48.5** | **45.2** | **0.704** | **0.660** | **0.509** |
| $\Delta_{baseline}$ | **+1.7** | **+2.4** | **+2.1** | **+4.7** | **+4.5** | **+3.8** | **+2.4** | **+3.2** | **+2.7** | **+0.5** | **+0.7** | **+0.6** | **+0.058** | **+0.094** | **+0.069** |
| **SparseMM** | 61.6 | 60.8 | 57.6 | **51.9** | **50.7** | 46.2 | 65.1 | 64.7 | 62.8 | 51.9 | 51.2 | 48.9 | 0.680 | 0.634 | 0.524 |
| + MixKV | **61.9** | **61.0** | **59.2** | 50.8 | 50.4 | **49.5** | **65.2** | **65.0** | **64.4** | 51.8 | **51.5** | **50.6** | **0.682** | **0.652** | **0.575** |
| $\Delta_{baseline}$ | **+0.3** | **+0.2** | **+1.6** | -1.1 | -0.3 | **+3.3** | **+0.1** | **+0.3** | **+1.6** | -0.1 | **+0.3** | **+1.7** | **+0.002** | **+0.018** | **+0.051** |
| *InternVL3-8B* | | | | | | | | | | | | | | | |
| *Full KV* | | 91.0 | | | 84.2 | | | 81.1 | | | 86.4 | | | 1.042 | |
| **SnapKV** | 89.2 | 85.4 | 75.7 | 80.6 | 69.0 | **53.1** | 80.4 | 78.2 | 71.9 | 86.2 | 84.6 | 79.8 | 1.009 | 0.901 | 0.734 |
| + MixKV | **89.4** | **86.2** | **76.3** | **81.9** | **71.1** | 52.3 | **80.9** | **78.8** | **72.9** | **86.3** | **84.8** | **80.7** | **1.029** | **0.949** | **0.753** |
| $\Delta_{baseline}$ | **+0.2** | **+0.8** | **+0.6** | **+1.3** | **+2.1** | -0.8 | **+0.5** | **+0.6** | **+1.0** | **+0.1** | **+0.2** | **+0.9** | **+0.020** | **+0.048** | **+0.019** |
| **PyramidKV** | 87.2 | 82.7 | 69.7 | 70.9 | 58.4 | **41.8** | 78.3 | 75.3 | 67.2 | 85.7 | 84.0 | 78.0 | 0.896 | 0.809 | 0.632 |
| + MixKV | **87.5** | **83.5** | **70.4** | **72.3** | **60.0** | 41.2 | **79.0** | **76.6** | **68.2** | **85.8** | **84.4** | **78.6** | **0.941** | **0.850** | **0.646** |
| $\Delta_{baseline}$ | **+0.3** | **+0.8** | **+0.7** | **+1.4** | **+1.6** | -0.6 | **+0.7** | **+1.3** | **+1.0** | **+0.1** | **+0.4** | **+0.6** | **+0.045** | **+0.041** | **+0.014** |
| **AdaKV** | 89.2 | 86.0 | 77.2 | 80.8 | 70.2 | **53.1** | 80.4 | 78.0 | 71.8 | 86.2 | 84.4 | 80.4 | 1.013 | 0.921 | 0.759 |
| + MixKV | **89.5** | **86.7** | **78.1** | **82.4** | **71.6** | 52.3 | **80.8** | **78.7** | **72.9** | 86.2 | **85.2** | **80.9** | **1.034** | **0.955** | **0.782** |
| $\Delta_{baseline}$ | **+0.3** | **+0.7** | **+0.9** | **+1.6** | **+1.4** | -0.8 | **+0.4** | **+0.7** | **+1.1** | **+0.0** | **+0.8** | **+0.5** | **+0.021** | **+0.034** | **+0.023** |
| *Qwen2-VL-7B-Instruct* | | | | | | | | | | | | | | | |
| *Full KV* | | 93.9 | | | 82.1 | | | 82.1 | | | 81.5 | | | 1.469 | |
| **SnapKV** | 88.0 | 80.1 | 66.5 | 77.3 | 71.9 | 62.4 | 80.3 | 77.5 | 69.9 | 81.3 | 79.6 | 75.5 | 1.360 | 1.142 | 0.794 |
| + MixKV | **90.5** | **82.6** | **67.9** | **79.3** | **75.4** | **66.0** | **81.9** | **80.6** | **72.5** | **81.6** | **81.2** | **77.6** | **1.470** | **1.342** | **0.878** |
| $\Delta_{baseline}$ | **+2.5** | **+2.5** | **+1.4** | **+2.0** | **+3.5** | **+3.6** | **+1.6** | **+3.1** | **+2.6** | **+0.3** | **+1.6** | **+2.1** | **+0.110** | **+0.200** | **+0.084** |
| **PyramidKV** | 81.7 | 74.0 | 59.9 | 74.5 | 67.9 | 56.8 | 78.3 | 74.6 | 65.3 | 81.1 | 79.2 | 73.5 | 1.115 | 0.951 | 0.569 |
| + MixKV | **84.0** | **76.3** | **60.8** | **76.6** | **72.6** | **58.4** | **80.4** | **77.1** | **67.0** | **81.3** | **80.7** | **75.5** | **1.348** | **1.119** | **0.633** |
| $\Delta_{baseline}$ | **+2.3** | **+2.3** | **+0.9** | **+2.1** | **+4.7** | **+1.6** | **+2.1** | **+2.5** | **+1.7** | **+0.2** | **+1.5** | **+2.0** | **+0.233** | **+0.168** | **+0.064** |
| **AdaKV** | 87.4 | 81.2 | 67.1 | 77.8 | 71.0 | 62.1 | 79.9 | 77.0 | 70.3 | 80.8 | 79.6 | 75.9 | 1.345 | 1.146 | 0.775 |
| + MixKV | **90.3** | **82.1** | **67.8** | **79.3** | **74.7** | **65.5** | **81.8** | **79.6** | **71.2** | **81.5** | **80.9** | **77.4** | **1.448** | **1.275** | **0.878** |
| $\Delta_{baseline}$ | **+2.9** | **+0.9** | **+0.7** | **+1.5** | **+3.7** | **+3.4** | **+1.9** | **+2.6** | **+0.9** | **+0.7** | **+1.3** | **+1.5** | **+0.103** | **+0.129** | **+0.103** |
| **SparseMM** | 93.5 | 91.5 | 84.9 | 81.2 | 79.0 | 74.3 | **82.0** | 81.6 | 77.3 | 82.0 | 81.5 | 80.1 | **1.482** | 1.430 | 1.038 |
| + MixKV | **93.8** | **92.7** | **86.4** | **82.0** | **81.0** | **77.1** | **82.0** | **82.0** | **80.9** | 81.6 | **81.8** | **81.4** | 1.480 | **1.459** | **1.303** |
| $\Delta_{baseline}$ | **+0.3** | **+1.2** | **+1.5** | **+0.8** | **+2.0** | **+2.8** | **+0.0** | **+0.4** | **+3.6** | -0.4 | **+0.3** | **+1.3** | -0.002 | **+0.029** | **+0.265** |

**Implementation Details.** We integrate MixKV with various KV compression methods, including SnapKV (Li et al., 2024b), PyramidKV (Cai et al., 2025b), AdaKV (Feng et al., 2025), and SparseMM (Wang et al., 2025b), across different KV cache budgets. Details are in Appendix A.3.

## 4.2 MAIN RESULTS

**Performance on Multi-modal Understanding Benchmarks.** Table 1 summarizes MixKV integrated with baseline methods across models and benchmarks, and highlights *three key advantages*: **(i) Universal improvements:** MixKV consistently enhances baselines across architectures, tasks, and budgets, supporting the benefit of mixing importance with diversity. **(ii) Broad applicability:** it benefits a wide range of paradigms, from simple baselines such as SnapKV to layer-wise or head-wise budget allocation methods (PyramidKV and SparseMM), by modifying only the evaluation function without changing the compression operator. **(iii) Model compatibility:** MixKV estimates head-wise redundancy on-the-fly per sample, avoiding offline statistics (as in SparseMM) and enabling a plug-and-play integration with existing LVLMs. Additional results in Table 7 and Table 6 further verify its robustness on larger LVLMs and MoE-based models, including InternVL3-38B (Zhu et al., 2025) and Qwen3-VL-30B-A3B-Instruct (Bai et al., 2025a).

**Performance on GUI Grounding Benchmarks.** Recent advancements in LVLMs demonstrate their capability to understand GUI scenarios (Cheng et al., 2024; Tang et al., 2025b;a), which rely on edge-side comprehension and necessitate KV compression to reduce storage demands. To this end,

Table 2: **Performance on ScreenSpot-v2 GUI grounding benchmark with Qwen2.5-VL-7B-Instruct.** "Full KV" refers to caching all KV pairs of the LLM (upper bound).

| Methods | Mobile Text | | Mobile Icon/Widget | | Desktop Text | | Desktop Icon/Widget | | Web Text | | Web Icon/Widget | | Average | |
|---|---|---|---|---|---|---|---|---|---|---|---|---|---|---|
| | 128 | 64 | 128 | 64 | 128 | 64 | 128 | 64 | 128 | 64 | 128 | 64 | 128 | 64 |
| **Qwen2.5-VL-7B-Instruct** | | | | | | | | | | | | | | |
| **Full KV** | 97.2 | | 87.7 | | 91.2 | | 77.1 | | 88.5 | | 82.3 | | 88.5 | |
| **SnapKV** | 65.5 | 28.6 | 78.7 | 53.1 | 86.1 | 57.2 | 74.3 | 57.1 | 76.9 | 46.6 | 74.4 | 49.8 | 75.3 | 46.9 |
| + MixKV | 86.6 | 35.5 | 85.3 | 60.2 | 87.1 | 71.1 | 75.0 | 65.7 | 85.0 | 53.0 | 76.4 | 56.2 | 83.3 | 54.9 |
| $\Delta_{baseline}$ | +21.1 | +6.9 | +6.6 | +7.1 | +1.0 | +13.9 | +0.7 | +8.6 | +8.1 | +6.4 | +2.0 | +6.4 | +7.9 | +8.0 |
| **PyramidKV** | 45.5 | 11.0 | 62.1 | 34.1 | 82.0 | 33.0 | 75.0 | 47.9 | 69.2 | 20.5 | 71.4 | 24.1 | 65.6 | 26.1 |
| + MixKV | 64.1 | 15.9 | 74.4 | 42.2 | 87.1 | 41.8 | 74.3 | 47.1 | 76.9 | 24.8 | 71.9 | 27.1 | 74.1 | 31.1 |
| $\Delta_{baseline}$ | +18.6 | +4.9 | +12.3 | +8.1 | +5.1 | +8.8 | -0.7 | -0.8 | +7.7 | +4.3 | +0.5 | +3.0 | +8.5 | +5.0 |
| **AdaKV** | 80.7 | 35.2 | 84.8 | 59.2 | 90.2 | 70.6 | 74.3 | 63.6 | 82.1 | 49.6 | 75.9 | 56.2 | 81.6 | 53.7 |
| + MixKV | 94.1 | 49.0 | 88.6 | 66.8 | 89.7 | 75.3 | 75.0 | 68.6 | 85.0 | 61.5 | 76.9 | 63.1 | 86.0 | 62.7 |
| $\Delta_{baseline}$ | +13.4 | +13.8 | +3.8 | +7.6 | -0.5 | +4.7 | +0.7 | +5.0 | +2.9 | +12.0 | +1.0 | +6.9 | +4.4 | +9.0 |

Table 3: **Performance on LongBench with Mistral-7B-Instruct-v0.2 and Llama-3.1-8B-Instruct.** "Full KV" refers to caching all KV pairs of the LLM (upper bound).

| Methods | Information Localization | | | | | | Information Aggregation | | | | | | | | | | Avg. |
|---|---|---|---|---|---|---|---|---|---|---|---|---|---|---|---|---|---|
| | Single-Doc QA | | | Multi-Doc QA | | | Summarization | | | Few-shot | | | Synthetic | | Code | | |
| | NrtvQA | Qasper | MF-en | HotpotQA | 2WikiMQA | Musique | GovReport | QMSum | MultiNews | TREC | TriviaQA | SAMSum | PCount | PRe | Lcc | RB-P | |
| **Mistral-7B-Instruct-v0.2** | | | | | | | | | | | | | | | | | |
| **Full KV** | 26.81 | 33.19 | 49.26 | 43.02 | 27.12 | 18.78 | 32.80 | 24.16 | 27.02 | 71.00 | 86.23 | 42.64 | 2.75 | 86.98 | 55.09 | 53.01 | 42.49 |
| **KV Cache Budget = 1024** | | | | | | | | | | | | | | | | | |
| **SnapKV** | 24.98 | 30.24 | 49.03 | 41.45 | 27.11 | 18.26 | 25.69 | 23.87 | 25.97 | 68.00 | 86.25 | 42.30 | 2.82 | 87.93 | 54.95 | 52.00 | 41.30 |
| + MixKV | 25.55 | 31.04 | 48.19 | 41.31 | 27.18 | 19.24 | 26.98 | 23.88 | 26.74 | 70.00 | 86.46 | 43.77 | 2.90 | 85.99 | 55.02 | 51.28 | 41.60 |
| $\Delta_{baseline}$ | +0.57 | +0.80 | -0.84 | -0.14 | +0.07 | +0.98 | +1.29 | +0.01 | +0.77 | +2.00 | +0.21 | +1.47 | +0.08 | -1.94 | +0.07 | -0.72 | +0.30 |
| **AdaKV** | 25.15 | 30.60 | 49.06 | 40.93 | 26.92 | 18.81 | 25.88 | 23.96 | 25.84 | 69.00 | 86.24 | 43.01 | 2.85 | 88.68 | 55.19 | 52.46 | 41.54 |
| + MixKV | 25.31 | 30.56 | 48.83 | 41.96 | 26.95 | 18.27 | 26.77 | 23.85 | 26.37 | 70.50 | 86.63 | 43.44 | 2.62 | 86.52 | 55.65 | 51.87 | 41.63 |
| $\Delta_{baseline}$ | +0.16 | -0.04 | -0.23 | +1.03 | +0.03 | -0.54 | +0.89 | -0.11 | +0.53 | +1.50 | +0.39 | +0.43 | -0.23 | -2.16 | +0.46 | -0.59 | +0.09 |
| **KV Cache Budget = 512** | | | | | | | | | | | | | | | | | |
| **SnapKV** | 23.69 | 27.71 | 49.16 | 39.70 | 25.44 | 17.38 | 23.31 | 23.28 | 24.20 | 66.00 | 86.17 | 41.54 | 3.24 | 86.29 | 53.71 | 51.19 | 40.13 |
| + MixKV | 23.56 | 28.19 | 48.96 | 40.36 | 25.86 | 17.34 | 24.63 | 23.36 | 25.32 | 66.00 | 86.23 | 42.25 | 3.02 | 87.66 | 53.87 | 51.40 | 40.50 |
| $\Delta_{baseline}$ | -0.13 | +0.48 | -0.20 | +0.66 | +0.42 | -0.04 | +1.32 | +0.08 | +1.12 | 0.00 | +0.06 | +0.71 | -0.22 | +1.37 | +0.16 | +0.21 | +0.37 |
| **AdaKV** | 24.35 | 27.33 | 48.76 | 40.07 | 26.38 | 17.97 | 23.73 | 23.51 | 24.31 | 67.50 | 86.38 | 42.53 | 3.06 | 86.65 | 53.90 | 51.57 | 40.50 |
| + MixKV | 24.26 | 28.39 | 48.90 | 40.86 | 26.33 | 17.07 | 24.63 | 23.32 | 25.41 | 69.00 | 86.51 | 42.67 | 3.07 | 86.44 | 54.46 | 51.69 | 40.81 |
| $\Delta_{baseline}$ | -0.09 | +1.06 | +0.14 | +0.79 | -0.05 | -0.90 | +0.90 | -0.19 | +1.10 | +1.50 | +0.13 | +0.14 | +0.01 | -0.21 | +0.56 | +0.12 | +0.31 |
| **Llama-3.1-8B-Instruct** | | | | | | | | | | | | | | | | | |
| **Full KV** | 30.22 | 45.37 | 55.80 | 55.97 | 45.00 | 31.26 | 35.12 | 25.38 | 27.20 | 72.50 | 91.64 | 43.57 | 9.41 | 99.50 | 62.88 | 56.43 | 49.20 |
| **KV Cache Budget = 1024** | | | | | | | | | | | | | | | | | |
| **SnapKV** | 27.10 | 43.91 | 55.07 | 55.60 | 45.17 | 30.47 | 27.84 | 24.44 | 25.75 | 69.00 | 91.89 | 42.69 | 9.44 | 99.50 | 62.49 | 56.30 | 48.86 |
| + MixKV | 27.50 | 44.19 | 55.42 | 55.82 | 45.40 | 30.65 | 28.83 | 24.75 | 26.26 | 70.00 | 91.62 | 42.88 | 8.96 | 99.50 | 62.69 | 56.41 | 49.30 |
| $\Delta_{baseline}$ | +0.40 | +0.28 | +0.35 | +0.22 | +0.23 | +0.18 | +0.99 | +0.31 | +0.51 | +1.00 | -0.27 | +0.19 | -0.48 | +0.00 | +0.20 | +0.11 | +0.44 |
| **AdaKV** | 28.16 | 43.98 | 54.68 | 56.14 | 45.19 | 30.30 | 28.35 | 24.80 | 26.11 | 72.50 | 91.72 | 42.48 | 8.74 | 99.50 | 62.94 | 56.51 | 49.27 |
| + MixKV | 27.98 | 44.28 | 55.03 | 56.03 | 45.58 | 30.55 | 29.06 | 24.58 | 26.70 | 72.50 | 91.42 | 43.37 | 9.46 | 99.50 | 62.65 | 56.97 | 49.37 |
| $\Delta_{baseline}$ | -0.18 | +0.30 | +0.35 | -0.11 | +0.39 | +0.25 | +0.71 | -0.22 | +0.59 | +0.00 | -0.30 | +0.89 | +0.72 | +0.00 | -0.29 | +0.46 | +0.10 |
| **KV Cache Budget = 512** | | | | | | | | | | | | | | | | | |
| **SnapKV** | 27.42 | 38.95 | 53.57 | 55.20 | 44.68 | 29.75 | 25.55 | 24.21 | 24.28 | 64.50 | 92.35 | 41.04 | 9.98 | 99.50 | 62.50 | 54.93 | 46.53 |
| + MixKV | 26.76 | 41.77 | 53.77 | 55.19 | 44.72 | 30.02 | 26.03 | 24.28 | 25.27 | 69.00 | 91.44 | 42.24 | 9.98 | 99.50 | 61.84 | 55.17 | 47.37 |
| $\Delta_{baseline}$ | -0.66 | +2.82 | +0.20 | -0.01 | +0.04 | +0.27 | +0.48 | +0.07 | +0.99 | +4.50 | -0.91 | +1.20 | +0.00 | +0.00 | -0.66 | +0.24 | +0.84 |
| **AdaKV** | 25.96 | 40.26 | 52.82 | 54.55 | 43.83 | 30.43 | 25.76 | 24.06 | 24.69 | 69.00 | 92.05 | 42.10 | 9.45 | 99.50 | 62.58 | 55.59 | 46.42 |
| + MixKV | 26.13 | 42.08 | 53.18 | 55.47 | 43.88 | 28.80 | 26.68 | 24.03 | 25.35 | 70.00 | 91.01 | 42.79 | 9.41 | 99.50 | 62.92 | 55.82 | 46.75 |
| $\Delta_{baseline}$ | +0.17 | +1.82 | +0.36 | +0.92 | +0.05 | -1.63 | +0.92 | -0.03 | +0.66 | +1.00 | -1.04 | +0.69 | -0.04 | +0.00 | +0.34 | +0.23 | +0.33 |

we further evaluate the fundamental GUI grounding capability on ScreenSpot-v2 using Qwen2.5-VL-7B-Instruct (Bai et al., 2025b). Table 2 shows that integrating MixKV with baseline importance-based compression methods yields significant GUI grounding performance improvements. Notably, with a budget of 128, SnapKV (Li et al., 2024b) improves average precision from 75.3% to 83.3%, achieving a performance boost of **+7.9%**. This further validates the effectiveness of MixKV and its potential for edge-side deployment of GUI agent models.

**Performance on Long-Context Text Benchmarks.** Table 3 evaluates the integration of MixKV with importance-based methods on Mistral-7B-Instruct-v0.2 Jiang et al. (2023) and Llama3.1-8B-Instruct (Grattafiori et al., 2024) using LongBench (Bai et al., 2024), demonstrating its applicability to long-context text tasks in LLMs. Overall, MixKV yields consistent gains in average performance, with more pronounced improvements under tighter KV budgets. We also observe intriguing patterns: in Information Aggregation tasks like Summarization, MixKV substantially enhances baselines by preserving diverse KV pairs, enabling better global information coverage essential for synthesis. However, in Information Localization tasks, occasional declines occur, likely because these require focused retrieval of local salient details, and introducing diversity may dilute attention in LLMs,

Table 4: **Ablation on `MixKV` metrics.** $s_{\text{imp}}^{\text{ex}}$: extrinsic importance only (baseline). $^{\dagger}W^{\text{head}}$: offline head weights (OCRBench-derived, sample-shared $\bar{r}_h^l$). $W^{\text{head}}$: online head weights (per-sample $\bar{r}_h^l$).

| Settings | DocVQA (%) | | | OCRBench (%) | | | TextVQA (%) | | | ChartQA (%) | | | TextCaps | | |
|---|---|---|---|---|---|---|---|---|---|---|---|---|---|---|---|
| | SnapKV | AdaKV | SparseMM | SnapKV | AdaKV | SparseMM | SnapKV | AdaKV | SparseMM | SnapKV | AdaKV | SparseMM | SnapKV | AdaKV | SparseMM |
| **LLaVA-NeXT-Mistral-7B, KV Cache Budget = 64** | | | | | | | | | | | | | | | |
| *Effects of Different Importance Metrics* | | | | | | | | | | | | | | | |
| $s_{\text{imp}}^{\text{ex}}$ (baseline) | 47.3 | 48.7 | 57.6 | 31.9 | 32.8 | 46.2 | 57.1 | 56.9 | 62.8 | 42.7 | 44.6 | 48.9 | 0.444 | 0.440 | 0.524 |
| $s_{\text{imp}}^{\text{ex}} + s_{\text{imp}}^{\text{in (KNorm)}}$ | 46.7 | 48.2 | 55.4 | 30.7 | 32.0 | 41.6 | 56.4 | 56.6 | 60.9 | 42.6 | 43.8 | 47.4 | 0.445 | 0.444 | 0.482 |
| $s_{\text{imp}}^{\text{ex}} + s_{\text{imp}}^{\text{in (VNorm)}}$ | 48.2 | 49.7 | 59.1 | 34.4 | 35.0 | 49.0 | 58.0 | 57.9 | 64.0 | 43.7 | 45.3 | 50.3 | 0.470 | 0.469 | 0.544 |
| $\Delta_{\text{baseline}}$ | +0.9 | +1.0 | +1.5 | +2.5 | +2.2 | +2.8 | +0.9 | +1.0 | +1.2 | +1.0 | +0.7 | +1.4 | +0.026 | +0.029 | +0.020 |
| *Effects of Different Mixing Strategies* | | | | | | | | | | | | | | | |
| $s_{\text{div}}$ | 34.2 | 35.8 | 51.8 | 28.4 | 28.8 | 43.9 | 54.5 | 55.2 | 63.4 | 32.3 | 34.2 | 48.1 | 0.487 | 0.504 | 0.534 |
| $s_{\text{imp}} + s_{\text{div}}$ | 48.8 | 50.6 | 59.1 | 36.1 | 36.1 | 49.5 | 59.8 | 59.1 | 64.4 | 43.5 | 45.4 | 50.9 | 0.516 | 0.504 | 0.573 |
| $\Delta_{\text{baseline}}$ | +1.5 | +2.1 | +1.6 | +4.2 | +3.8 | +3.3 | +3.0 | +2.7 | +1.6 | +0.9 | +0.6 | +1.7 | +0.070 | +0.069 | +0.051 |
| **Qwen2-VL-7B-Instruct, KV Cache Budget = 64** | | | | | | | | | | | | | | | |
| *Effects of Different Importance Metrics* | | | | | | | | | | | | | | | |
| $s_{\text{imp}}^{\text{ex}}$ (baseline) | 66.5 | 67.1 | 84.9 | 62.4 | 62.1 | 74.3 | 69.9 | 70.3 | 77.3 | 75.5 | 75.9 | 80.1 | 0.794 | 0.775 | 1.038 |
| $s_{\text{imp}}^{\text{ex}} + s_{\text{imp}}^{\text{in (KNorm)}}$ | 66.1 | 66.5 | 81.2 | 56.2 | 56.0 | 68.8 | 67.9 | 67.2 | 72.6 | 71.4 | 72.8 | 76.3 | 0.766 | 0.769 | 0.927 |
| $s_{\text{imp}}^{\text{ex}} + s_{\text{imp}}^{\text{in (VNorm)}}$ | 67.2 | 67.4 | 84.7 | 64.9 | 64.4 | 75.5 | 71.5 | 70.3 | 79.7 | 77.3 | 77.4 | 80.8 | 0.862 | 0.854 | 1.259 |
| $\Delta_{\text{baseline}}$ | +0.7 | +0.3 | -0.2 | +2.5 | +2.3 | +1.2 | +1.6 | +0.0 | +2.4 | +1.8 | +1.5 | +0.7 | +0.068 | +0.079 | +0.221 |
| *Effects of Different Mixing Strategies* | | | | | | | | | | | | | | | |
| $s_{\text{div}}$ | 44.0 | 44.3 | 60.4 | 50.7 | 50.3 | 68.4 | 59.8 | 59.7 | 78.4 | 64.7 | 65.2 | 78.5 | 0.739 | 0.711 | 1.113 |
| $s_{\text{imp}} + s_{\text{div}}$ | 67.6 | 67.6 | 86.3 | 65.0 | 63.6 | 76.9 | 72.2 | 70.6 | 80.8 | 76.6 | 77.0 | 81.4 | 0.905 | 0.869 | 1.291 |
| $^{\dagger}W^{\text{head}}(s_{\text{imp}} + s_{\text{div}})$ | 67.7 | 67.7 | 86.2 | 66.1 | 65.2 | 76.8 | 72.4 | 71.1 | 80.9 | 77.5 | 77.3 | 81.1 | 0.922 | 0.873 | 1.301 |
| $W^{\text{head}}(s_{\text{imp}} + s_{\text{div}})$ | 67.9 | 67.8 | 86.4 | 66.0 | 65.5 | 77.1 | 72.5 | 71.2 | 80.9 | 77.6 | 77.4 | 81.5 | 0.916 | 0.879 | 1.303 |
| $\Delta_{\text{baseline}}$ | +1.4 | +0.7 | +1.5 | +3.6 | +3.4 | +2.8 | +2.6 | +0.9 | +3.6 | +2.1 | +1.5 | +1.4 | +0.122 | +0.104 | +0.265 |

Figure 4: **Efficiency comparisons of total latency and peak memory.** For a context length of 32,000, "Full KV" refers to caching the entire sequence, whereas KV compression strategies employ a budget of 64. The upper part is total time, while the lower part is peak memory.

where head-wise semantic redundancy is inherently lower than in LVLMs (Figure 1). This highlights the task-dependent benefits of balancing importance and diversity.

## 4.3 ABLATION STUDIES AND ANALYSIS

**Ablation of Different Metrics in `MixKV`.** Table 4 systematically evaluates three components: (a) importance scores combining extrinsic ($s_{\text{imp}}^{\text{ex}}$) and two intrinsic measures ($s_{\text{imp}}^{\text{in (KNorm)}}$, $s_{\text{imp}}^{\text{in (VNorm)}}$); (b) diversity scores ($s_{\text{div}}$); and (c) head-wise adaptive mixing ($W^{\text{head}}$). We select LLaVA-NeXT-Mistral-7B and Qwen2-VL-7B with different architectures to ensure generalizability.

For importance metrics, we evaluate combinations of extrinsic importance with intrinsic measures: $s_{\text{imp}}^{\text{ex}} + s_{\text{imp}}^{\text{in (KNorm)}}$ and $s_{\text{imp}}^{\text{ex}} + s_{\text{imp}}^{\text{in (VNorm)}}$. Results show that jointly assessing extrinsic importance of keys and intrinsic importance of values provides consistent performance gains across scenarios. However, $s_{\text{imp}}^{\text{ex}} + s_{\text{imp}}^{\text{in (KNorm)}}$ underperforms as KNorm focuses on key magnitude, potentially misaligning with value-driven attention dynamics in LVLMs and leading to suboptimal KV pair retention. Conversely, $s_{\text{imp}}^{\text{ex}} + s_{\text{imp}}^{\text{in (VNorm)}}$ proves more effective, with VNorm better capturing value contributions to multi-modal attention, enhancing task relevance and compression efficiency.

Beyond importance, we validate the effects of mixing importance with diversity. Results demonstrate that relying solely on diversity ($s_{\text{div}}$) is inadequate, leading to performance degradation due to the disruption of original semantic information. Furthermore, jointly incorporating importance and diversity through $s_{\text{imp}} + s_{\text{div}}$ achieves notable performance improvements across diverse models and benchmarks. We observe that applying $\bar{r}_h^l$, computed offline on OCRBench (Liu et al., 2024c), to other samples and benchmarks ($^{\dagger}W^{\text{head}}(s_{\text{imp}} + s_{\text{div}})$), further enhances performance compared to the direct combination $s_{\text{imp}} + s_{\text{div}}$, as it accounts for varying redundancy levels across heads and adaptively adjusts importance and diversity weights. Advancing this, computing $\bar{r}_h^l$ per sam-

ple for $W^{\text{head}}(s_{\text{imp}} + s_{\text{div}})$ yields additional performance gains, since this approach allows `MixKV` to adaptively tune importance and diversity weights based on the characteristics of each processed sample, optimizing the mixing effect for superior performance. Moreover, experiments reveal that the inference costs of $W^{\text{head}}(s_{\text{imp}} + s_{\text{div}})$ and $^{\dagger}W^{\text{head}}(s_{\text{imp}} + s_{\text{div}})$ remain comparable, as the low computational cost of $\bar{r}_h^l$ calculation (Equation 4) does not affect the inference efficiency.

**Efficiency Analysis of `MixKV`.** Figure 4 compares the inference latency and peak memory consumption of the base model (Qwen2-VL-7B-Instruct with full KV cache), various standalone KV cache compression baselines, and our `MixKV` framework applied on top of these baselines. As expected, all compression methods reduce both latency and memory compared to the full KV cache baseline, validating their role in enhancing LVLM inference efficiency. Crucially, integrating `MixKV` with existing baseline compression methods yields substantial performance gains *without degrading their original efficiency*. This is because the overhead introduced by `MixKV`, primarily the lightweight mixing operation, scales linearly with the KV sequence length $T$ and remains negligible (*e.g.*, less than 1% increase in latency) compared to the cost of the underlying compression method and the overall generation process. Collectively, the findings in Table 4 and Figure 4 affirm that `MixKV` achieves an exceptional balance between task performance and inference efficiency. More additional efficiency analysis is presented in Appendix A.5.

## 5 CONCLUSION

In this work, we analyze KV pair characteristics in LVLMs and identify two critical distinctions: LVLMs exhibit significantly higher semantic redundancy than LLMs, and attention heads demonstrate varying redundancy patterns. Based on these insights, we propose `MixKV`, which jointly optimizes importance and diversity for KV cache compression. `MixKV` quantifies semantic similarity within each attention head and adaptively balances importance and diversity weights, prioritizing diversity in high redundancy heads while emphasizing importance in low redundancy heads. Extensive experiments across multiple models and benchmarks confirm that `MixKV` consistently enhances existing compression methods while maintaining inference efficiency.

## ACKNOWLEDGEMENTS

This research was supported by the Shanghai Science and Technology Program (Grant No. 25ZR1402278).

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

# A APPENDIX

## A.1 ADDITIONAL EXPLANATION OF FIGURE 1-3

**Figure 1:** To investigate the semantic redundancy patterns in KV caches across model architectures, we visualize the head-wise average cosine similarity of key vectors in Qwen2-7B (Yang et al., 2024) (an LLM) and Qwen2-VL-7B (Wang et al., 2024) (an LVLM), using representative samples from LongBench (Bai et al., 2024) and OCRBench (Liu et al., 2024c). This comparison reveals that LVLMs exhibit substantially higher intra-head semantic redundancy than LLMs.

**Figure 2:** To quantify the variability of redundancy across attention heads within LVLMs, we randomly select 100 samples from each benchmark and compute the average cosine similarity of key vectors per head in Qwen2-VL-7B (Wang et al., 2024) and LLaVA-NeXT-Mistral-7B (Liu et al., 2024b). The results demonstrate significant head-wise heterogeneity in semantic redundancy, motivating our adaptive compression strategy.

**Figure 3:** To qualitatively compare the semantic coverage of different KV selection strategies, we randomly select one sample from TextVQA (Singh et al., 2019) and apply PCA to project the key vectors from a representative attention head (layer 23, head 3 in the LLM) into a two-dimensional space. This head exhibits moderate semantic redundancy, making it an ideal case to illustrate how different compression strategies balance information retention. We visualize the distributions of keys retained under three settings: full KV cache, SnapKV (Li et al., 2024b), and our `MixKV`. In Figure 3, `MixKV` preserves a broader and more diverse set of key vectors compared to SnapKV.

## A.2 MORE DISCUSSIONS ON REDUNDANCY DIFFERENCES

To further validate the two types of "redundancy differences" introduced in Section 1, Figure 5 visualizes the head-wise KV cache redundancy of Qwen2 (Yang et al., 2024) and Qwen2-VL (Wang et al., 2024) when processing pure-text and vision-language inputs. The pure-text and vision-language results are obtained by averaging over 100 samples randomly drawn from LongBench (Bai et al., 2024) and TextVQA (Singh et al., 2019), respectively. For pure-text inputs, Qwen2 and Qwen2-VL exhibit highly similar redundancy patterns across heads, suggesting that the architectural difference alone does not account for the redundancy gap we observe.

From this visualization, we obtain *two key findings:* **(I) Vision-Language Redundancy Differences.** When Qwen2-VL processes vision-language inputs, the semantic redundancy of its KV cache is substantially higher than for pure-text inputs, which is consistent with our analysis in the main text. Notably, for some heads (*e.g.*, Layer 29, Heads 0 and 1), the redundancy on vision-language data is more than twice that on text-only data, reflecting the inherently redundant nature of visual signals, whereas textual tokens tend to be more semantically diverse. **(II) Head-wise Redundancy Differences.** For both pure-text and vision-language data, different heads exhibit markedly different redundancy levels, and their overall patterns are highly similar: a head that is relatively more redundant on text remains relatively more redundant on vision-language inputs. We hypothesize that this is because different heads focus on different types of information: some heads primarily attend to local patterns and therefore exhibit higher semantic redundancy, while others capture more global information and consequently show much lower redundancy.

## A.3 DETAILED EXPERIMENT SETTINGS

**Model Details**  We introduce more details of LVLMs used for evaluation in the main text:

- **LLaVA-NeXT** (Liu et al., 2024b) improves upon LLaVA (Liu et al., 2023; 2024a) by supporting higher resolutions (4 × more pixels) and multiple aspect ratios using an AnyRes technique. It employs a simple linear projector to connect vision features to the LLM, enabling efficient multi-modal processing with unified interleaving of visual and text tokens.

- **InternVL3** (Zhu et al., 2025) is an advanced vision-language model in the InternVL series (Chen et al., 2024c;b), following the "ViT-MLP-LLM" architecture. It features native multi-modal pre-training for superior performance in multi-modal tasks, with dynamic resolution handling and efficient alignment between vision and language components.

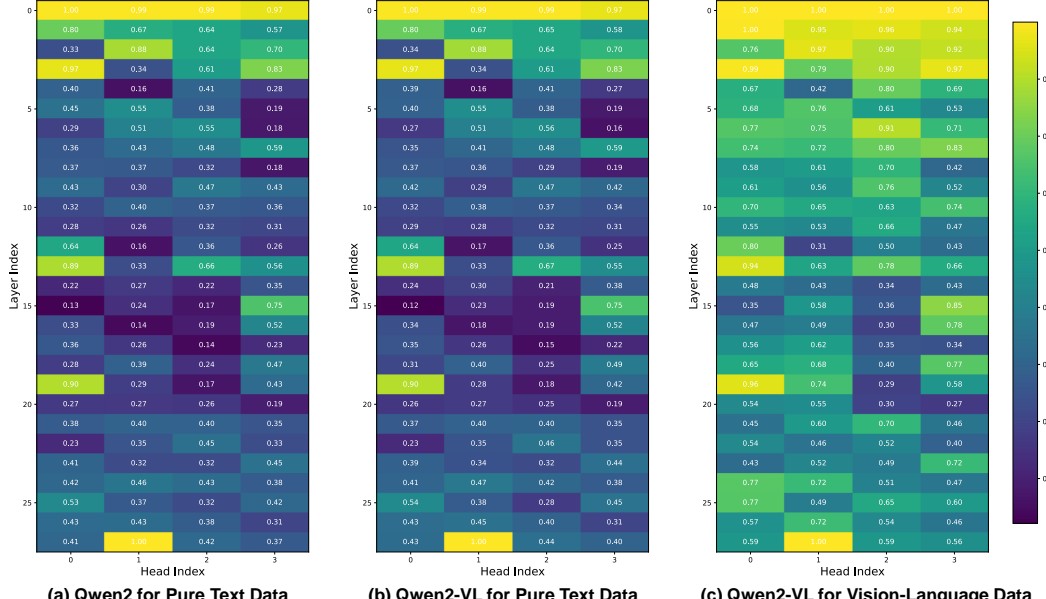

Figure 5: **Visualization of KV cache redundancy of Qwen2 and Qwen2-VL on different data types.** The number on each head is the average cosine similarity of the KV cache within that head.

- **Qwen2-VL** (Wang et al., 2024) introduces Naive Dynamic Resolution to adaptively convert frames of any resolution into visual tokens. It utilizes multi-modal Rotary Position Embedding (M-RoPE) within a unified image-and-video processing paradigm, enabling the handling of long videos for high-quality QA, dialogue, and content creation.

- **Mistral-7B** (Jiang et al., 2023) is a dense transformer-based LLM with 7B parameters. It adopts Grouped-Query Attention (GQA) and Sliding Window Attention (SWA), improving inference efficiency while supporting long-context understanding. Despite its compact size, it delivers competitive performance across reasoning, coding, and dialogue tasks.

- **Llama 3** (Grattafiori et al., 2024) represents the latest generation of Llama models, offering a family of parameter scales (e.g., 8B, 70B). It leverages large-scale pretraining with optimized data curation and advanced instruction tuning, resulting in strong performance on benchmarks covering reasoning, knowledge-intensive tasks, and multi-turn dialogue.

**Benchmark Details**  We provide detailed introductions of benchmarks used in the main text:

- **DocVQA** (Mathew et al., 2021) is a visual question answering benchmark on document images, focusing on extracting and reasoning over information from scanned documents.

- **OCRBench** (Liu et al., 2024c) evaluates OCR abilities, testing models on diverse text extraction tasks from images with varying fonts, layouts, and noise levels.

- **TextVQA** (Singh et al., 2019) requires models to read and reason about text in images to answer questions, emphasizing multi-modal integration of visual and textual information.

- **ChartQA** (Masry et al., 2022) assesses visual question answering on charts and graphs, requiring models to interpret data visualizations and answer related questions accurately.

- **TextCaps** (Sidorov et al., 2020) is a captioning benchmark for text-containing images, focusing on descriptions that accurately incorporate and describe textual content in scenes.

- **ScreenSpot-v2** (Wu et al., 2024) is a GUI grounding benchmark that evaluates a model's ability to locate and identify specific UI elements (*e.g.*, icons, text buttons) within screenshots across diverse platforms including mobile, desktop, and web interfaces.

- **LongBench** (Bai et al., 2024) evaluates long-context language understanding, with tasks testing models' handling of extended sequences across reasoning and comprehension.

Table 5: **Performance of integrating HeadKV into `MixKV` on LongBench and LooGLE benchmarks using Mistral-7B-Instruct-v0.2.**

| Methods | Single-Doc QA | | | Multi-Doc QA | | | Avg. | Long Dependency QA | | | | Avg. |
|---------|---------|--------|-------|----------|----------|---------|-------|--------|----------------|----------|-------------|-------|
| | NartvQA | Qasper | MF-en | HotpotQA | 2WikiMQA | Musique | | Doc.QA | Info. Retrieval | Timeline | Computation | |
| Mistral-7B-Instruct-v0.2 | | | | | | | | | | | | |
| Full KV | 26.63 | 32.99 | 49.34 | 42.77 | 27.35 | 18.78 | 32.98 | 12.17 | 15.52 | 0.49 | 10.03 | 9.55 |
| KV Cache Budget = 1024 | | | | | | | | | | | | |
| HeadKV | 25.88 | **31.28** | **50.54** | 40.61 | 27.57 | 18.80 | 32.45 | 11.93 | **14.87** | 0.49 | **9.56** | **9.21** |
| + `MixKV` | **26.26** | 31.20 | 50.07 | **40.99** | **27.88** | **19.93** | **32.72** | **12.06** | 14.73 | **0.50** | 9.33 | 9.16 |
| $\Delta_{\text{baseline}}$ | +0.38 | -0.08 | -0.47 | +0.38 | +0.31 | +1.13 | +0.27 | +0.13 | -0.14 | +0.01 | -0.23 | -0.05 |
| KV Cache Budget = 128 | | | | | | | | | | | | |
| HeadKV | 24.34 | 26.60 | 48.55 | 40.69 | 25.97 | 15.34 | 30.25 | 10.48 | 12.72 | 0.53 | 10.04 | 8.44 |
| + `MixKV` | **24.39** | **27.70** | **49.85** | **42.48** | **27.21** | **15.40** | **31.17** | **10.62** | **13.08** | **0.73** | **10.31** | **8.69** |
| $\Delta_{\text{baseline}}$ | +0.05 | +1.10 | +1.30 | +1.79 | +1.24 | +0.06 | +0.92 | +0.14 | +0.36 | +0.20 | +0.27 | +0.25 |

**Baseline Details** We provide a detailed introduction to the baseline importance-based KV cache compression methods used in the main text:

- **SnapKV** (Li et al., 2024b) clusters important KV positions based on attention patterns observed from an initial window of tokens, enabling efficient KV cache compression by retaining only the most relevant clusters while maintaining high generation quality and reducing memory usage during inference.

- **PyramidKV** (Cai et al., 2025b) dynamically adjusts KV cache sizes across different LLM layers in a pyramidal manner, allocating more KV cache budget to lower layers for foundational information and less to higher layers for refined processing, based on information priority to optimize compression and performance.

- **AdaKV** (Feng et al., 2025) adaptively allocates eviction budgets across attention heads of LLMs by evaluating head-specific contributions, providing a plug-and-play solution for KV cache compression that significantly reduces memory footprint while preserving model performance in generative inference tasks.

- **SparseMM** (Wang et al., 2025b) exploits sparsity patterns in visual attention heads of multi-modal models, assigning asymmetric KV cache budgets based on head importance for visual tokens, enabling modality-aware compression that effectively reduces storage requirements in vision-language models without sacrificing accuracy.

- **KNorm** (Devoto et al., 2024) compresses KV cache using the $\ell_2$ norm of key embeddings, keeping low-norm keys that correlate with high attention scores.

- **VNorm** (Kim et al., 2025) ranks tokens by the $\ell_2$ norm of their value embeddings to preserve semantically salient information.

## A.4 ADDITIONAL EXPERIMENTS

**Performance of `MixKV` with HeadKV for Long-Context Understanding.** Table 5 further applies HeadKV (Fu et al., 2025) within our `MixKV` framework to validate its generality. Experimental results show that integrating HeadKV into `MixKV` can improve pure-text long-context understanding. In particular, under the extreme compression setting (*i.e.*, Budget=128), we observe **consistent and substantial gains** across all tasks. This suggests that, when the KV cache budget is highly constrained, it is crucial to preserve KV entries that are both important and diverse in order to maintain the long-context understanding ability of the baseline models.

**Performance of `MixKV` on Larger LVLMs for Multi-Modal Understanding.** Table 6 reports the results of integrating `MixKV` with baseline KV cache compression methods on the larger LVLM InternVL3-38B (Zhu et al., 2025) to evaluate its effectiveness for multi-modal understanding. Experimental results show that `MixKV` **consistently improves all baseline methods** across benchmarks and KV cache budgets, demonstrating its robustness on larger LVLMs and further highlighting its practical value for real-world deployment.

**Performance of `MixKV` on MoE-based LVLMs for Multi-Modal Understanding.** Table 7 further presents the results of integrating `MixKV` with SnapKV (Li et al., 2024b) on the MoE-based LVLM Qwen3-VL-30B-A3B-Instruct to evaluate its performance on recent MoE-style LVLM architectures.

Table 6: **Performance of Applying `MixKV` to InternVL3-38B.**

| Methods | DocVQA (%) | | OCRBench (%) | | TextVQA (%) | | ChartQA (%) | | TextCaps | |
|---|---|---|---|---|---|---|---|---|---|---|
| | 128 | 64 | 128 | 64 | 128 | 64 | 128 | 64 | 128 | 64 |
| InternVL3-38B | | | | | | | | | | |
| **Full KV** | 93.5 | | 85.9 | | 83.8 | | 88.6 | | 0.953 | |
| **SnapKV** | 87.5 | 85.2 | 77.8 | 64.3 | 82.0 | 78.5 | 87.5 | 85.2 | 0.932 | 0.822 |
| + `MixKV` | **92.1** | **86.9** | **79.3** | **65.8** | **82.8** | **79.4** | **88.2** | **85.8** | **0.959** | **0.859** |
| $\Delta_{\text{baseline}}$ | **+4.6** | **+1.7** | **+1.5** | **+1.5** | **+0.8** | **+0.9** | **+0.7** | **+0.6** | **+0.027** | **+0.037** |
| **AdaKV** | 92.0 | 87.6 | 79.6 | 67.8 | 82.0 | 79.3 | 87.4 | 85.3 | 0.940 | 0.841 |
| + `MixKV` | **92.3** | **88.5** | **81.1** | **69.2** | **82.9** | **80.2** | **88.2** | **86.0** | **0.961** | **0.859** |
| $\Delta_{\text{baseline}}$ | **+0.3** | **+0.9** | **+1.5** | **+1.4** | **+0.9** | **+0.9** | **+0.8** | **+0.7** | **+0.021** | **+0.018** |

Table 7: **Performance of Applying `MixKV` to Qwen3-VL-30B-A3B-Instruct.**

| Methods | DocVQA (%) | | OCRBench (%) | | TextVQA (%) | | ChartQA (%) | | TextCaps | |
|---|---|---|---|---|---|---|---|---|---|---|
| | 128 | 64 | 128 | 64 | 128 | 64 | 128 | 64 | 128 | 64 |
| Qwen3-VL-30B-A3B-Instruct | | | | | | | | | | |
| **Full KV** | 94.5 | | 84.0 | | 83.5 | | 85.1 | | 0.287 | |
| **SnapKV** | 91.9 | 83.8 | 71.0 | 55.2 | 75.3 | 75.3 | 83.8 | 79.8 | 0.314 | 0.272 |
| + `MixKV` | **93.2** | **86.2** | **80.7** | **68.8** | **80.8** | **79.7** | **84.5** | **80.8** | **0.411** | **0.349** |
| $\Delta_{\text{baseline}}$ | **+1.3** | **+2.4** | **+9.7** | **+13.6** | **+5.5** | **+4.4** | **+0.7** | **+1.0** | **+0.097** | **+0.077** |

Experimental results show that `MixKV` can significantly boost SnapKV across various benchmarks; in particular, on OCRBench, it brings a **13.6%** improvement under the strict Budget=64 setting. These results demonstrate the strong effectiveness of `MixKV` on MoE-based LVLMs and further highlight its potential for improving the efficiency of future LVLM inference.

## A.5 MORE EFFICIENCY ANALYSIS OF `MixKV`

Figure 6 further compares the total inference latency and peak GPU memory consumption across different KV cache compression settings. We observe that integrating `MixKV` with baseline methods (*e.g.*, SnapKV, AdaKV) incurs negligible overhead, typically less than 1% increase in latency and no measurable rise in peak memory, while consistently achieving the same level of computational efficiency as the underlying baselines. This confirms that `MixKV` preserves the original inference speed and memory footprint of the compression method it enhances, making it a truly plug-and-play efficiency-preserving framework.

## A.6 ALGORITHM DETAILS OF `MixKV`

Algorithm 1 outlines the workflow of our `MixKV` framework, seamlessly integrated with SnapKV (Li et al., 2024b) to enhance its performance.

## A.7 LIMITATIONS AND FUTURE WORK.

Our study is conducted on models up to the 8B parameter scale. Future work should validate if the observed head-wise redundancy patterns and the effectiveness of `MixKV` generalize to significantly larger models (*e.g.*, 70B+). This would be a crucial step for broader applicability.

## A.8 THE USE OF LARGE LANGUAGE MODELS

In this study, we solely employ LLM-based language polishing to refine sentence fluency and correct grammatical errors, without altering the technical content or experimental data of the paper.

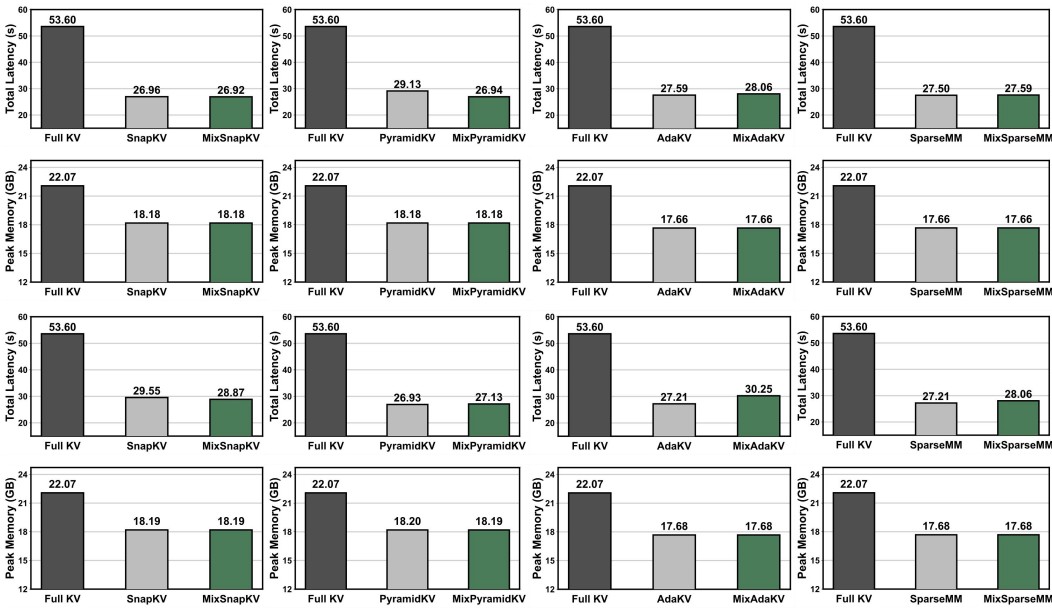

Figure 6: **More efficiency comparisons of total latency and peak memory.** The top two rows of bars correspond to a budget of 128, while the bottom two rows correspond to a budget of 256.

---

**Algorithm 1** `MixKV` with SnapKV for KV Cache Compression

---

1: **Input:** Key-value pairs $\mathbf{K}_h^l, \mathbf{V}_h^l$, query values $\mathbf{Q}$, total memory budget $B$.
2: **Output:** Compressed KV pairs $\hat{\mathbf{K}}_h^l, \hat{\mathbf{V}}_h^l$
3: Let $B' = B - |\text{window}|$ denote the adjusted budget for non-window KV pairs.
4: **for** each layer $l$ and head $h$ **do**
5:     **Step 1: Compute Importance and Diversity Scores**
6:     Compute the intrinsic importance using VNorm, normalize, and scale:

$$s_{\text{scaled},i}^{\text{in}} = \frac{\|\mathbf{V}_{h,i}^l\|_2 - \min_j(\|\mathbf{V}_{h,j}^l\|_2)}{\max_j(\|\mathbf{V}_{h,j}^l\|_2) - \min_j(\|\mathbf{V}_{h,j}^l\|_2) + \epsilon} \cdot \frac{\bar{s}_{\text{imp}}^{\text{ex}}}{\bar{s}_{\text{norm}}^{\text{in}} + \epsilon}, \quad i = 1 \text{ to } T$$

7:     Compute the extrinsic importance as average attention scores:

$$s_{\text{imp},i}^{\text{ex}} = \frac{1}{|\text{window}|} \sum_{j \in \text{window}} \text{Attention}(\mathbf{Q}_j, \mathbf{K}_i), \quad i = 1 \text{ to } T$$

8:     Combine importance scores:

$$s_{\text{imp},i} = s_{\text{imp},i}^{\text{ex}} + s_{\text{scaled},i}^{\text{in}}, \quad i = 1 \text{ to } T$$

9:     Normalize keys and compute diversity scores:

$$s_i^{\text{div}} = -\frac{\mathbf{K}_{h,i}^l}{\|\mathbf{K}_{h,i}^l\|} \cdot \frac{1}{T} \sum_{i=1}^T \frac{\mathbf{K}_{h,i}^l}{\|\mathbf{K}_{h,i}^l\|}, \quad i = 1 \text{ to } T$$

10:     **Step 2: Head-wise Adaptive Mixing**
11:     Quantify redundancy and compute comprehensive scores:

$$\bar{r}_h^l = \frac{T^2 \left\| \frac{1}{T} \sum_{i=1}^T \frac{\mathbf{K}_{h,i}^l}{\|\mathbf{K}_{h,i}^l\|} \right\|_2^2 - T}{T(T-1)}$$

$$s_{\text{scaled},i}^{\text{div}} = \frac{s_i^{\text{div}} - \min_j(s_j^{\text{div}})}{\max_j(s_j^{\text{div}}) - \min_j(s_j^{\text{div}}) + \epsilon} \cdot \frac{\bar{s}_{\text{imp}}}{\bar{\bar{s}}_{\text{div}} + \epsilon}, \quad i = 1 \text{ to } T$$

$$s_i^{\text{comp}} = (1 - \bar{r}_h^l) \cdot s_{\text{imp},i} + \bar{r}_h^l \cdot s_{\text{scaled},i}^{\text{div}}, \quad i = 1 \text{ to } T$$

12:     Select the top-$B'$ KV pairs based on comprehensive scores:

$$\hat{\mathbf{K}}_h^l, \hat{\mathbf{V}}_h^l = \text{TopB}(\mathbf{K}_h^l[\text{exclude window}], \mathbf{V}_h^l[\text{exclude window}], \{s_i^{\text{comp}}\}_{i=1}^T)$$

13: **end for**
14: **Return:** Compressed KV cache $C = \{(\hat{\mathbf{K}}_h^l, \hat{\mathbf{V}}_h^l)\}_{l,h}$

---

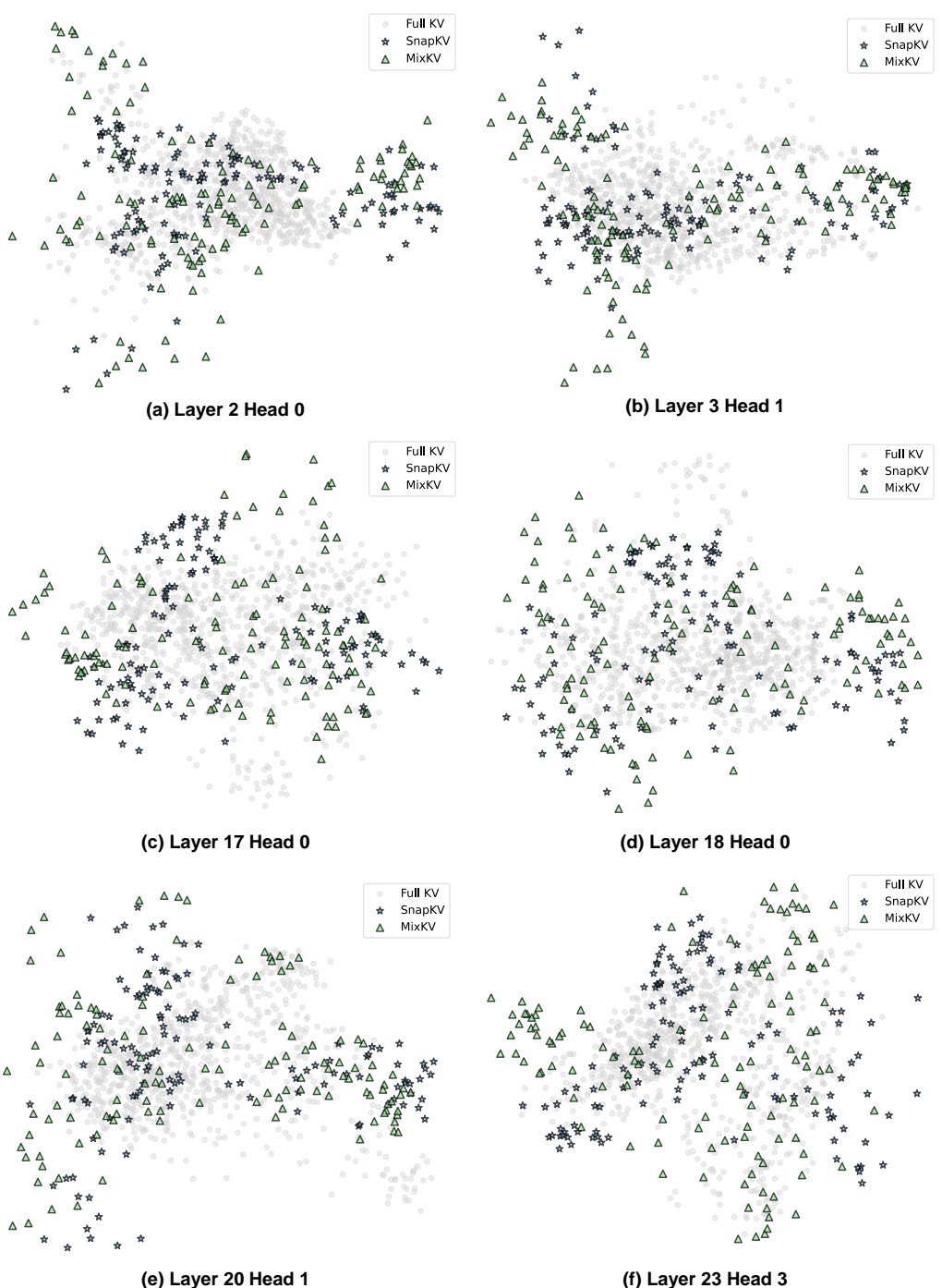

Figure 7: **More t-SNE visualization of KV cache distributions under different settings.**

