# OpenReview forum: "Mixing Importance with Diversity: Joint Optimization for KV Cache Compression in Large Vision-Language Models"
_ICLR.cc/2026/Conference — ICLR 2026 Poster_

### Official Review · Reviewer_nMmG · 2025-10-18

**Soundness:** 3
**Presentation:** 3
**Contribution:** 3
**Rating:** 6
**Confidence:** 4

**Summary:**

This paper introduces MixKV, a simple, plug-and-play KV-cache compression method for (multi-modal) LLMs that mixes standard importance scores with a diversity metric at the per-head level. It estimates head-level redundancy via the off-diagonal average similarity of normalized keys and uses this statistic to adapt the diversity–importance mix, complementing attention- and value-norm–based importance signals. The approach is lightweight—requiring no architectural changes—and, across multiple LVLMs/LLMs under tight KV budgets, consistently outperforms importance-only baselines. Experiments on both multimodal understanding and language modeling tasks further demonstrate its effectiveness.

**Strengths:**

The paper is well written and easy to follow. It convincingly diagnoses the limitations of importance-only KV compression and motivates a diversity component, positioning the work clearly against prior baselines. The evaluation is comprehensive—spanning multiple models and benchmarks with budget sweeps and ablations (e.g., choice of intrinsic signal, effect of mixing)—and it substantiates the central claims.

**Weaknesses:**

The paper notes that LVLMs exhibit much higher key similarity. However, the proposed approach appears general and seemingly applicable to LLMs as well. Which components, if any, are specifically designed for LVLMs, and what prevents a straightforward application to text-only LLMs?

**Questions:**

1. The study is thorough at the 7B dense scale, but scalability to larger backbones and MoE architectures remains unclear. Results on a larger dense LVLM (e.g., 34B–72B) and on an MoE model such as Qwen/Qwen3-VL-30B-A3B-Instruct would help substantiate generalization.

2. In Section 4.3 and Table 4, the “online head weights” setup is under-specified. Define what “online” means (e.g., per-token, no lookahead) and detail the procedure for computing and applying these weights—i.e., which signals are used and at what step/layer they are measured.

3. The authors only report results on image understanding benchmarks. How does the proposed method perform on video understanding tasks, such as Video-MME [1]? Including evaluations on video benchmarks would help demonstrate the method’s generality across temporal visual–language settings.

4. Table 3 presents results on long-context understanding tasks, but the reported LLMs are not specifically optimized for extended context lengths. To make the comparison more meaningful, it would be valuable to include models that have been explicitly tuned for long-context inference, such as Qwen/Qwen3-4B-Instruct-2507. This would better reflect the effectiveness of the proposed method under realistic long-context settings.

Reference:

[1] Video-mme: The first-ever comprehensive evaluation benchmark of multi-modal llms in video analysis. arXiv 2024.

---

> ### Author Response · Authors · 2025-11-23
> **Response to Reviewer nMmG (part 1)**
>
> We sincerely appreciate your recognition of our work, and we believe your suggestions have significantly improved the quality of our paper. Below, we provide point-by-point responses to your comments.
>
> **W. The paper notes that LVLMs exhibit much higher key similarity. However, the proposed approach appears general and seemingly applicable to LLMs as well. Which components, if any, are specifically designed for LVLMs, and what prevents a straightforward application to text-only LLMs?**
>
> **A1.**   We thank the reviewer for this insightful question. Our method is indeed **motivated by LVLM-specific observations, but it is not restricted to LVLMs**.
>
> Our analysis first shows that, when processing vision–language inputs, LVLMs exhibit (i) substantially higher semantic redundancy in their KV cache than text-only LLMs, and (ii) highly heterogeneous redundancy across attention heads. These findings are supported by our main-text analysis and the updated Figure 5 in Appendix A.2, where Qwen2-VL shows much stronger and more uneven head-wise redundancy than its pure-text counterpart Qwen2. Under this LVLM-specific redundancy structure, purely importance-based KV compression (e.g., SnapKV, AdaKV, PyramidKV, SparseMM) fails to preserve complementary information from seemingly “redundant” heads, leading to large gaps from Full KV (Figure 3). This motivates MixKV: we explicitly quantify semantic redundancy and use it to balance importance and diversity when selecting KV entries.
>
> At the same time, **MixKV is not limited to vision-language tasks**: in this paper, we also evaluate it on long-context pure-text benchmarks. Specifically, the submitted manuscript shows that MixKV brings consistent gains over SnapKV and AdaKV on LongBench for Mistral-7B and Llama-3.1-8B (Table 3), and the revised version further integrates HeadKV into the MixKV framework and evaluates on LongBench and LooGLE (10 tasks). The new results (Table 5, Appendix A.4) demonstrate that MixKV consistently improves HeadKV under constrained KV budgets (e.g., Budget = 128), indicating that the redundancy structures exploited by MixKV, while more pronounced in LVLMs, are also present and beneficial to leverage in text-only models.

---

> > ### Author Response · Authors · 2025-11-23
> > **Response to Reviewer nMmG (part 2)**
> >
> > **Q1. The study is thorough at the 7B dense scale, but scalability to larger backbones and MoE architectures remains unclear. Results on a larger dense LVLM (e.g., 34B–72B) and on an MoE model such as Qwen/Qwen3-VL-30B-A3B-Instruct would help substantiate generalization.**
> >
> >
> > **A2.** We thank the reviewer for this valuable suggestion and fully agree that evaluating larger backbones and MoE architectures is important for assessing generalization. We would like to clarify that Qwen3-VL-30B-A3B-Instruct was released after the ICLR submission deadline, so it was not included in our original set of baseline LVLMs. In the revised manuscript, we address this concern by **adding experiments on both a larger dense LVLM and an MoE-based LVLM**.
> >
> > Specifically, Appendix A.4 now reports results on (i) a larger dense LVLM, **InternVL3-38B**, and (ii) the MoE LVLM **Qwen3-VL-30B-A3B-Instruct** (Tables 6 and 7, highlighted in red). In both settings, integrating MixKV with KV compression baselines consistently improves performance across multi-modal benchmarks and KV cache budgets, demonstrating the robustness and generality of MixKV.
> >
> > | **Methods**          | **DocVQA** |         | **OCRBench** |         | **TextVQA** |         | **ChartQA** |         | **TextCaps**    |         |
> > |------------------|------------|---------|--------------|---------|-------------|---------|-------------|---------|-------------|---------|
> > |                  | 128        | 64      | 128          | 64      | 128         | 64      | 128         | 64      | 128         | 64      |
> > | **InternVL3-38B** |            |         |              |         |             |         |             |         |             |         |
> > | Full KV          | 93.5       |         | 85.9         |         | 83.8        |         | 88.6        |         | 0.953       |         |
> > | SnapKV           | 87.5       | 85.2    | 77.8         | 64.3    | 82.0        | 78.5    | 87.5        | 85.2    | 0.932       | 0.822   |
> > | + MixKV          | **92.1**       | **86.9**    | **79.3**         | **65.8**    | **82.8**        | **79.4**    | **88.2**        | **85.8**    | **0.959**       | **0.859**   |
> > | Δ        | +4.6       | +1.7    | +1.5         | +1.5    | +0.8        | +0.9    | +0.7        | +0.6    | +0.027      | +0.037  |
> > | AdaKV            | 92.0       | 87.6    | 79.6         | 67.8    | 82.0        | 79.3    | 87.4        | 85.3    | 0.940       | 0.841   |
> > | + MixKV          | **92.3**       | **88.5**    | **81.1**         | **69.2**    | **82.9**        | **80.2**    | **88.2**        | **86.0**    | **0.961**       | **0.859**   |
> > | Δ        | +0.3       | +0.9    | +1.5         | +1.4    | +0.9        | +0.9    | +0.8        | +0.7    | +0.021      | +0.018  |
> >
> > | Methods          | DocVQA |         | OCRBench |         | TextVQA |         | ChartQA |         | TextCaps    |         |
> > |------------------|------------|---------|--------------|---------|-------------|---------|-------------|---------|-------------|---------|
> > |                  | 128        | 64      | 128          | 64      | 128         | 64      | 128         | 64      | 128         | 64      |
> > | **Qwen3-VL-30B-A3B-Instruct** |            |         |              |         |             |         |             |         |             |         |
> > | Full KV          | 94.5       |         | 84.0         |         | 83.5        |         | 85.1        |         | 0.287       |         |
> > | SnapKV           | 91.9       | 83.8    | 71.0         | 55.2    | 75.3        | 75.3    | 83.8        | 79.8    | 0.314       | 0.272   |
> > | + MixKV          | **93.2**       | **86.2**    | **80.7**         | **68.8**    | **80.8**        | **79.7**    | **84.5**        | **80.8**    | **0.411**       | **0.349**   |
> > | Δ        | +1.3       | +2.4    | +9.7         | +13.6   | +5.5        | +4.4    | +0.7        | +1.0    | +0.097      | +0.077  |

---

> > > ### Author Response · Authors · 2025-11-23
> > > **Response to Reviewer nMmG (part 3)**
> > >
> > > **Q2. In Section 4.3 and Table 4, the “online head weights” setup is under-specified. Define what “online” means (e.g., per-token, no lookahead) and detail the procedure for computing and applying these weights—i.e., which signals are used and at what step/layer they are measured.**
> > >
> > > **A3.** We thank the reviewer for pointing this out. In Section 4.3, the “online head weights” setup corresponds to the head-wise adaptive mixing weights defined in Section 3. Concretely, for each input sample during inference, we compute the redundancy score \(\bar{r}_h^l\) for every head \(h\) at layer \(l\) using Eqs. (3)–(4), based solely on the KV cache of the **current sample** (no lookahead over other examples). These scores are computed from the cosine similarity between key vectors within each head, and are then used in Eq. (5) as the head-wise weights that balance the importance-based score and the diversity-based score for that sample. This is what we refer to as “online head weights”: the weights are computed **per sample**, on-the-fly, from the currently processed KV cache.
> > >
> > > For comparison, the “offline head weights” in Table 4 are obtained by first running the model on OCRBench and computing \(\bar{r}_h^l\) for each sample, then averaging these values over the dataset to obtain a **fixed** redundancy prior for each head and layer. At inference time, this offline variant uses these pre-computed average head weights instead of recomputing them per sample. This setting is analogous to SparseMM, which also relies on offline-estimated head scores.
> > >
> > > Our ablation in Table 4 shows that the online variant yields better performance than the offline one, while incurring almost identical runtime: the redundancy scores in Eqs. (3)–(4) are implemented with linear-time operations over the KV cache and add negligible overhead in practice. Based on this observation, we adopt online head weights as the default configuration of MixKV.
> > >
> > > **Q3. The authors only report results on image understanding benchmarks. How does the proposed method perform on video understanding tasks, such as Video-MME [1]? Including evaluations on video benchmarks would help demonstrate the method’s generality across temporal visual–language settings.**
> > >
> > >
> > > **A4.** We thank the reviewer for this helpful suggestion. We agree that video understanding requires the model to process longer and more complex multimodal contexts, and we have therefore added results for MixKV on a video understanding benchmark. We first clarify that VideoMME is a multiple-choice QA benchmark where an LVLM outputs only a single token to select an answer, which makes it unsuitable for evaluating the effect of KV cache compression on generative quality. Instead, we adopt VATEX, a large-scale, high-quality video captioning dataset that assesses a model’s video caption generation ability and thus better reflects the impact of KV cache compression. Experimental results on VATEX show that MixKV improves the performance of baseline KV cache compression methods, demonstrating its robustness in video scenarios with longer multi-modal context.
> > >
> > > | Methods        | Bleu\_4 |         |         | CIDEr   |         |         | ROUGE\_L |         |         |
> > > |----------------|---------|---------|---------|---------|---------|---------|----------|---------|---------|
> > > |                | 512     | 256     | 128     | 512     | 256     | 128     | 512      | 256     | 128     |
> > > | SnapKV         | 0.0013  | 0.0010  | 0.0000  | 0.0008  | 0.0006  | 0.0001  | 0.3453   | 0.3404  | 0.3610  |
> > > | +MixKV         | **0.0017**| **0.0011**| 0.0000  | **0.0013**| **0.0008**| **0.0001**| **0.3502**| **0.3445**| **0.3632**|
> > > | Δ              | +0.0004 | +0.0001 | +0.0000 | +0.0005 | +0.0002 | +0.0000 | +0.0049  | +0.0041 | +0.0022 |
> > > | AdaKV          | 0.0019  | 0.0000  | 0.0006  | 0.0010  | 0.0006  | 0.0003  | 0.3435   | 0.3396  | 0.3558  |
> > > | +MixKV         | **0.0020**| **0.0011**| **0.0009**| **0.0012**| **0.0009**| **0.0004**| **0.3463**| **0.3450**| **0.3571**|
> > > | Δ              | +0.0001 | +0.0011 | +0.0003 | +0.0002 | +0.0003 | +0.0001 | +0.0028  | +0.0054 | +0.0013 |

---

> > > > ### Author Response · Authors · 2025-11-23
> > > > **Response to Reviewer nMmG (part 4)**
> > > >
> > > > **Q4. Table 3 presents results on long-context understanding tasks, but the reported LLMs are not specifically optimized for extended context lengths. To make the comparison more meaningful, it would be valuable to include models that have been explicitly tuned for long-context inference, such as Qwen/Qwen3-4B-Instruct-2507. This would better reflect the effectiveness of the proposed method under realistic long-context settings.**
> > > >
> > > > **A5.** We thank the reviewer for this valuable suggestion. Our primary focus is on KV cache compression for LVLMs, so the main experiments are centered on multi-modal understanding. To address the reviewer’s concern more directly, in the revised manuscript we additionally evaluate MixKV on a long-context–tuned LVLM. Specifically, we include experiments on **Qwen3-VL-30B-A3B-Instruct**, which is explicitly optimized for long-context inference, and report the results in Appendix A.4 (**Table 7**, highlighted in red). When integrating MixKV with the baseline SnapKV on this model, we observe consistent improvements across multi-modal benchmarks and KV cache budgets; in particular, under the strict Budget = 64 setting on OCRBench, MixKV brings a **13.6-point** performance gain over SnapKV. These results provide direct evidence that MixKV remains effective under realistic long-context settings with models that are specifically tuned for extended context.

---

> > > > > ### Author Response · Authors · 2025-11-27
> > > > > **Kind Reminder Regarding Rebuttal**
> > > > >
> > > > > Dear Reviewer nMmG,
> > > > >
> > > > > Thank you sincerely for your valuable feedback. In response to your suggestions, we have updated our experimental results and uploaded a revised version of the manuscript.
> > > > >
> > > > > With the rebuttal deadline approaching in about one week, we would like to kindly ask: have your concerns or questions been fully addressed? If any point remains unclear or you need further information, please do not hesitate to let us know.
> > > > >
> > > > > Best regards,
> > > > > The Authors

---

### Official Review · Reviewer_vSGZ · 2025-10-31

**Soundness:** 2
**Presentation:** 3
**Contribution:** 2
**Rating:** 4
**Confidence:** 4

**Summary:**

The paper introduces MixKV, a KV cache compression method for large vision-language models (LVLMs) that combines importance and diversity to optimize memory usage and maintain performance. By adapting to head-wise redundancy, MixKV improves compression efficiency and achieves significant performance gains on multiple benchmarks.

**Strengths:**

The key advantage of MixKV is its comprehensive benchmarking across various tasks and models. It demonstrates consistent performance improvements across multiple multi-modal and text understanding benchmarks, including DocVQA, TextVQA, and LongBench, as well as GUI Grounding tasks.

**Weaknesses:**

1. I believe the paper spends unnecessary length discussing **modality-specific redundancy differences** and **head-wise redundancy differences**, as these concepts have already been well-established in previous works, such as **MMinference** and **VisionZip**. For instance, the **head-wise redundancy** can be directly observed in **MMinference's Fig. 1**, making it redundant to claim this in such detail. Additionally, I find it questionable to use **text on Qwen2** and **image on Qwen2-VL** to illustrate **modality-specific redundancy differences**. Normally, one would expect to examine both text and image together on a **Qwen2-VL** model to properly demonstrate this kind of redundancy, as that's where multi-modal data is processed simultaneously.

2. MixKV essentially merges two well-established paradigms: **importance-based** selection (FastV,ZipVL) and **diversity-based** retention (VisionZip). This type of fusion has already been explored in various contexts, such as in **attention mechanisms** and **token selection** techniques. The novelty here lies mainly in **applying this fusion to KV cache compression**, rather than introducing a fundamentally new idea.

3. The paper mainly tests on **images** or **multi-view data** and doesn't address the challenges of **long-context sequences**, like those found in **long video** tasks. Testing on a **20k context** in long video benchmarks would be essential to evaluate how well **MixKV** handles large, complex data with **long-range dependencies** and more diverse redundancy patterns, which might stress-test its memory compression capabilities.

**Questions:**

See weaknesses.

---

> ### Author Response · Authors · 2025-11-23
> **Response to Reviewer vSGZ (part 1)**
>
> We sincerely thank you for your constructive questions regarding our work; your feedback has greatly helped us improve the quality of our paper. Below, we provide point-by-point responses to your comments.
>
> **W1 (a). I believe the paper spends unnecessary length discussing modality-specific redundancy differences and head-wise redundancy differences, as these concepts have already been well-established in previous works, such as MMinference and VisionZip. For instance, the head-wise redundancy can be directly observed in MMinference's Fig. 1, making it redundant to claim this in such detail.**
>
> **A1 (a).** We thank the reviewer for raising this concern. We respectfully clarify that MMInference and VisionZip study a **different perspective** of redundancy from ours. In MMInference, Figure 1 does not analyze head-wise redundancy, and we suspect the reviewer may be referring to Figure 2 instead. However, Figure 2 in MMInference visualizes **sparse attention patterns** across modalities and positions, where the metric is the attention weight distribution. Similarly, VisionZip focuses on **visual token sparsity/redundancy** at the vision encoder output, again characterized by attention weights, and shows that LVLMs tend to attend to only a small subset of visual tokens.
>
>   In contrast, our Figures 1 and 2 analyze the **semantic redundancy of the KV cache** in LVLMs, where the metric is the cosine similarity between KV vectors within each head, **not the sparsity of attention weights**. Based on this semantic KV-level analysis, our introduction draws two empirical findings: (i) KV pairs in LVLMs exhibit substantially higher semantic redundancy than in LLMs; (ii) KV pairs in LVLMs show heterogeneous degrees of semantic redundancy across attention heads. These conclusions are fundamentally different from MMInference’s observations about modality-dependent sparse attention patterns, and from VisionZip’s focus on visual token sparsity and the fact that VLMs attend to only a few visual tokens.
>
>   We believe this analysis is necessary rather than redundant: prior KV cache compression works (e.g., SnapKV, AdaKV, PyramidKV, SparseMM) mainly emphasize **importance** and do not account for the **heterogeneous head-wise semantic redundancy** we uncover in LVLM KV cache. Our study shows that keeping only important KV cache can cause substantial information loss compared to the full KV cache (Figure 3), which directly motivates the design of MixKV to jointly consider importance and diversity.
>
>   Moreover, our analysis is **not limited to vision-language tasks**. In the revised manuscript, we further show that similar heterogeneous head-wise redundancy also appears in long-context **pure-text** settings, and that MixKV consistently improves baseline compression methods (e.g., SnapKV, AdaKV, HeadKV) on pure-text long-context tasks (Table 3/5). In contrast, MMInference and VisionZip focus exclusively on vision-language tasks. Our results therefore complement these works by revealing and exploiting a different, KV-level semantic redundancy structure that is present in both LVLMs and long-context LLMs.
>
>
> **W1 (b). Additionally, I find it questionable to use text on Qwen2 and image on Qwen2-VL to illustrate modality-specific redundancy differences. Normally, one would expect to examine both text and image together on a Qwen2-VL model to properly demonstrate this kind of redundancy, as that's where multi-modal data is processed simultaneously.**
>
> **A1 (b).**   We thank the reviewer for this helpful suggestion. In the revised version, we update **Figure 5 in Appendix A.2** by adding a pure LLM baseline (Qwen2) processing only text, alongside its LVLM counterpart (Qwen2-VL) processing both pure-text and vision-language inputs. Concretely, we sample 100 cases from LongBench (pure-text) and 100 cases from TextVQA (vision-language, where Qwen2-VL receives both image and text jointly) and compute head-wise KV semantic redundancy. The updated figure shows that Qwen2-VL exhibits higher semantic redundancy on vision-language inputs than on pure-text inputs, and that different heads have heterogeneous redundancy levels, which is consistent with our previously reported “Vision-Language Redundancy Differences” and “Head-wise Redundancy Differences”. We also clarify in the revised manuscript that Qwen2-VL is always fed image–text pairs in the vision-language setting, while Qwen2 is used only as a pure-text LLM baseline; more analysis is provided in **Appendix A.2**.

---

> > ### Author Response · Authors · 2025-11-23
> > **Response to Reviewer vSGZ (part 2)**
> >
> > **W2. MixKV essentially merges two well-established paradigms: importance-based selection (FastV,ZipVL) and diversity-based retention (VisionZip). This type of fusion has already been explored in various contexts, such as in attention mechanisms and token selection techniques. The novelty here lies mainly in applying this fusion to KV cache compression, rather than introducing a fundamentally new idea.**
> >
> > **A2.**   We thank the reviewer for this concern. We would like to clarify that our contribution goes beyond simply “applying the fusion of importance and diversity to KV cache compression” in a generic way. First, our work is explicitly targeted at KV cache compression for LVLMs, and is grounded in **new empirical findings** about KV pairs that, to the best of our knowledge, have not been systematically documented before. In the Introduction and the updated Appendix A.2, we identify two key characteristics of LVLM KV cache: (i) Vision-Language Redundancy Differences – KV pairs in LVLMs show substantially higher semantic redundancy when processing vision–language inputs than in text-only LLMs; and (ii) Head-wise Redundancy Differences – there exists **heterogeneous head-wise semantic redundancy** across attention heads, a phenomenon we find to be present in both LLMs and LVLMs. These head-wise redundancy patterns are different in nature from the token-level importance/diversity trade-offs explored in FastV, ZipVL, and VisionZip, which operate on attention weights or visual tokens rather than on the semantic structure of KV cache.
> >
> >  Building on these observations, MixKV is proposed not as a single heuristic, but as **a general framework** that can be seamlessly integrated into a wide range of existing importance-based KV compression methods (e.g., SnapKV, AdaKV, PyramidKV, SparseMM, HeadKV). Across multiple LVLMs and LLMs and a variety of benchmarks, MixKV consistently improves these baselines. In contrast, methods like FastV, ZipVL, and VisionZip are **token-centric** and cannot be directly applied to pure-text long-context understanding, and our new experiments (see our response to **Reviewer eFa6 Q2**) show that they suffer from significant degradation on visual text understanding tasks. By comparison, our MixKV does not discard the visual tokens themselves and thus avoids this failure mode. Furthermore, our rebuttal experiments demonstrate that combining token compression (FastV/VisionZip) with MixKV can improve their performance, while reducing both context length and KV storage. Thus, we believe these empirical findings and the demonstrated ability of MixKV to (i) exploit KV-specific redundancy, (ii) generalize across LVLMs and LLMs, and (iii) complement token-compression and architecture-level techniques, constitute a solid and meaningful contribution beyond simply reusing an existing “importance + diversity” idea.

---

> ### Author Response · Authors · 2025-11-23
> **Response to Reviewer vSGZ (part 3)**
>
> **W3. The paper mainly tests on images or multi-view data and doesn't address the challenges of long-context sequences, like those found in long video tasks. Testing on a 20k context in long video benchmarks would be essential to evaluate how well MixKV handles large, complex data with long-range dependencies and more diverse redundancy patterns, which might stress-test its memory compression capabilities.**
>
> **A3.** We thank the reviewer for highlighting the importance of evaluating MixKV on longer and more complex multimodal sequences. We agree that video tasks involve substantially longer context windows, stronger long-range dependencies, and more diverse redundancy patterns than image-based tasks. To address this concern, we have added experiments on the VATEX [1] video captioning benchmark—a large-scale, high-quality dataset that requires the model to process long video frame sequences and generate coherent captions. Using Qwen2-VL-7B as the baseline LVLM, we integrate MixKV into existing KV cache compression methods under different KV cache budgets (512/256/128) and observe consistent improvements across BLEU, ROUGE-L, and CIDEr metrics. These results demonstrate that MixKV remains effective even in long-sequence video scenarios, supporting its robustness on large and complex multi-modal inputs.
>
> | Methods        | Bleu\_4 |         |         | CIDEr   |         |         | ROUGE\_L |         |         |
> |----------------|---------|---------|---------|---------|---------|---------|----------|---------|---------|
> |                | 512     | 256     | 128     | 512     | 256     | 128     | 512      | 256     | 128     |
> | SnapKV         | 0.0013  | 0.0010  | 0.0000  | 0.0008  | 0.0006  | 0.0001  | 0.3453   | 0.3404  | 0.3610  |
> | +MixKV         | **0.0017**| **0.0011**| 0.0000  | **0.0013**| **0.0008**| **0.0001**| **0.3502**| **0.3445**| **0.3632**|
> | Δ              | +0.0004 | +0.0001 | +0.0000 | +0.0005 | +0.0002 | +0.0000 | +0.0049  | +0.0041 | +0.0022 |
> | AdaKV          | 0.0019  | 0.0000  | 0.0006  | 0.0010  | 0.0006  | 0.0003  | 0.3435   | 0.3396  | 0.3558  |
> | +MixKV         | **0.0020**| **0.0011**| **0.0009**| **0.0012**| **0.0009**| **0.0004**| **0.3463**| **0.3450**| **0.3571**|
> | Δ              | +0.0001 | +0.0011 | +0.0003 | +0.0002 | +0.0003 | +0.0001 | +0.0028  | +0.0054 | +0.0013 |
>
> In addition, MixKV has already shown strong performance on long-context pure-text tasks. As reported in the submitted manuscript (**Table 3**), MixKV improves SnapKV and AdaKV on LongBench (16 tasks), which includes multiple long-range–dependency tasks. In the revised manuscript, we further integrate HeadKV into the MixKV framework and evaluate it on LongBench and LooGLE (10 tasks, including Long Dependency QA). The new results (**Table 5, Appendix A.4**) show that MixKV consistently improves HeadKV, particularly under constrained KV budgets (e.g., Budget = 128). Taken together, the video experiments and long-context text experiments demonstrate that MixKV can effectively handle large, complex sequences with long-range dependencies and heterogeneous redundancy patterns, beyond the image or multi-view settings evaluated in the main paper.
>
> [1] Vatex: A large-scale, high-quality multilingual dataset for video-and-language research. CVPR, 2019.

---

> > ### Author Response · Authors · 2025-11-27
> > **Kind Reminder Regarding Rebuttal**
> >
> > Dear Reviewer vSGZ,
> >
> > Thank you sincerely for your insightful comments. Your feedback has greatly helped us improve the quality of our paper.
> >
> > As the rebuttal deadline is approaching in about one week, we would like to kindly check whether your concerns or questions have been adequately addressed. If any point remains unclear or you would like further discussion, please feel free to let us know.
> >
> > Best regards,
> > The Authors

---

### Official Review · Reviewer_eFa6 · 2025-11-01

**Soundness:** 4
**Presentation:** 4
**Contribution:** 3
**Rating:** 8
**Confidence:** 4

**Summary:**

This work introduces a novel KV cache compression method tailored for VLMs. The paper begins by presenting two core observations, supported by visualizations: (1) KV pairs in LVLMs exhibit substantially higher semantic redundancy than in text-only LLMs, and (2) KV pairs in VLMs show varying degrees of semantic redundancy across attention heads in the LLM. Motivated by these observations, the authors propose a new compression scheme named MixKV. The central idea is to perform head-wise KV cache compression by simultaneously considering both the semantic redundancy and the diversity of the tokens. The paper is evaluated across multiple distinct tasks and compared against several baselines, demonstrating consistent performance advantages under identical settings."

**Strengths:**

1.  The paper effectively identifies a key problem through insightful feature analysis and subsequently proposes a targeted solution. The motivation is well-grounded, the methodology is appropriate, and the results are solid and convincing.
2.  Although the proposed KV cache compression scheme is designed for VLMs, its effectiveness is also validated on text-only tasks. This demonstrates the method's strong generalizability and versatility.
3.  The experimental evaluation is thorough. The method demonstrates consistent performance gains across various tasks and when compared against multiple baseline methods.
4.  The problem addressed by this paper is of high practical significance, and the proposed solution has strong potential for real-world application in resource-constrained environments.

**Weaknesses:**

1.  No significant flaws were identified.
2.  However, the paper could be further strengthened by including an analysis of the persistent performance gap that remains when compared to the full KV cache.
3.  Additionally, a broader discussion that horizontally situates the proposed method among other KV cache (or GPU memory) saving techniques would be valuable. While direct experimental comparisons are not strictly necessary, a qualitative discussion of the trade-offs relative to other approaches would enhance the paper's context.

**Questions:**

1. What about more text tasks?
2. There is still a noticeable performance degradation when compared to the full KV cache. Can the authors include a discussion comparing the proposed method with other KV cache optimization techniques, such as MLA or similar approaches?

---

> ### Author Response · Authors · 2025-11-23
> **Response to Reviewer eFa6 (part 1)**
>
> Thank you for your positive evaluation. Your insightful questions have helped us significantly improve the quality of our paper. Below, we provide a point-by-point response to your comments.
>
> **W1. No significant flaws were identified.**
>
> **A1.** Thank you for your recognition of our work.
>
>
> **W2. However, the paper could be further strengthened by including an analysis of the persistent performance gap that remains when compared to the full KV cache.**
>
>
> **A2.** We thank the reviewer for this helpful suggestion. In the revised manuscript, we have updated the Full KV results in **Table 1**. As shown in the updated table, when the KV cache budget decreases, all compression methods, including MixKV, exhibit a noticeable performance drop compared to the Full KV setting. This is expected, since any fixed-budget KV compression inevitably discards part of the contextual information, which weakens long-range interactions and makes it harder for the model to recover fine-grained cues that the full KV cache can still access. Nevertheless, by encouraging the retained KV cache to be both important and diverse under a limited budget, MixKV still brings **consistent improvements** over the baseline compression methods, which supports the soundness of our design.
>
>
> **W3. Additionally, a broader discussion that horizontally situates the proposed method among other KV cache (or GPU memory) saving techniques would be valuable. While direct experimental comparisons are not strictly necessary, a qualitative discussion of the trade-offs relative to other approaches would enhance the paper's context.**
>
> **A3.** We thank the reviewer for this constructive suggestion. In the revised manuscript, we have **expanded Section 2** (Related Work) with a broader discussion of long-context optimization and GPU-memory–saving techniques (highlighted in red). Specifically, we now summarize **three complementary perspectives** frequently adopted in prior work: **(i)** *efficient computational architectures* (e.g., sparse attention, linear attention, and SSMs), **(ii)** *model-centric compression* (e.g., network pruning, model quantization, and knowledge distillation), and **(iii)** *data-centric compression* (e.g., token compression, KV cache compression, and KV cache quantization). These directions address long-context processing from **orthogonal** angles, each with its own trade-offs in accuracy, memory footprint, and implementation complexity. In general, the first two categories often rely on retraining or finetuning LVLMs, which can be data- and computation-intensive, whereas the third category is typically training-free and can be applied in a plug-and-play manner.
>
> Our method falls under the data-centric perspective and focuses on exploiting semantic redundancy in the KV cache to improve efficiency without modifying model architectures or training procedures. MixKV is complementary to most existing techniques and can be integrated alongside them. In addition, we provide experiments that combine MixKV with other long-context optimization methods; please refer to our response to **Q2** and the newly added tables in **Appendix A.4** for detailed results. We believe this expanded discussion provides the broader context requested by the reviewer and clarifies the positioning of MixKV within the long-context inference literature.

---

> ### Author Response · Authors · 2025-11-23
> **Response to Reviewer eFa6 (part 2)**
>
> **Q1. What about more text tasks?**
>
> **A4.**   We thank the reviewer for this suggestion. Our MixKV framework is **primarily designed for LVLMs**, where it adapts to head-wise semantic redundancy and selectively balances diversity and importance when compressing KV pairs. To examine whether these benefits transfer to pure-text LLMs, we have already included experiments on LongBench in the submitted manuscript (**Table 3**) by integrating MixKV with baseline compression methods. LongBench covers **16 sub-tasks**, including both Information Localization and Information Aggregation, and we evaluate two LLMs under two KV cache budgets.
>
> In addition, in the revised version we further add experiments in **Appendix A.4, Table 5**, where MixKV is combined with HeadKV on long-context pure-text tasks (LongBench and LooGLE, covering **10 sub-tasks**). The results consistently show that MixKV improves performance across these pure-text benchmarks, demonstrating its robustness beyond LVLMs and further highlighting its generalizability.
>
> **Q2. There is still a noticeable performance degradation when compared to the full KV cache. Can the authors include a discussion comparing the proposed method with other KV cache optimization techniques, such as MLA or similar approaches?**
>
> **A5.** We thank the reviewer for this helpful comment. As is common in KV cache compression, any method operating under a strict KV budget inevitably incurs a performance gap to the Full KV setting. Even so, our results show that MixKV consistently improves strong baselines (e.g., SnapKV, AdaKV, SparseMM) across diverse LVLMs and LLMs (Tables 1/2/3/5/6/7), while maintaining comparable end-to-end latency (Figures 4/6). This indicates that MixKV achieves a more favorable performance–efficiency trade-off than importance-only compression, especially under aggressive budgets.
>
> Regarding other KV cache optimization techniques, we view MixKV as largely complementary rather than competing. The baseline models we evaluate (LVLMs and LLMs) already adopt **Grouped-Query Attention (GQA)**, which reduces KV memory at the architecture level; MixKV further compresses the stored KV cache in a training-free, data-centric manner.
>
> We also compare MixKV with **token-centric compression** (FastV [1], VisionZip [2]) on Qwen2-VL-7B. Compressing visual tokens to CT = 128 severely harms visual-text understanding, especially on OCR-heavy benchmarks, as token compression removes visual information. In contrast, KV compression (SnapKV and MixKV) preserves visual tokens and avoids this issue. Moreover, applying moderate token compression (CT = 256/512) followed by MixKV allows us to reduce context length (improving prefill efficiency) and KV memory (improving decoding efficiency), while mitigating the performance drop of token-only compression.
>
>
> | **Methods**                                     | **DocVQA** | **OCRBench** | **TextVQA** | **ChartQA** |
> |----------------------------------------------|--------|----------|---------|---------|
> | FastV (CT=128)                               | 45.5   | 64.0     | 71.8    | 55.2    |
> | VisionZip (CT=128)                           | 49.2   | 63.2     | 70.6    | 66.1    |
> | SnapKV (KV=128)                              | 80.1   | 71.9     | 77.5    | 79.6    |
> | MixSnapKV (KV=128)                           | 82.6   | 75.4     | 80.6    | 81.2    |
>
> | **Methods**                                      | **DocVQA** | **OCRBench** | **TextVQA** | **ChartQA** |
> |----------------------------------------------|--------|----------|---------|---------|
> | FastV+MixSnapKV (CT=512, KV=128)             | 64.7   | 71.8     | 77.7    | 60.9    |
> | FastV+MixSnapKV (CT=256, KV=128)             | 76.3   | 74.9     | 80.1    | 71.1    |
> | VisionZip+MixSnapKV (CT=256, KV=128)         | 68.9   | 71.2     | 77.0    | 67.4    |
> | VisionZip+MixSnapKV (CT=512, KV=128)         | 78.0   | 74.1     | 80.3    | 72.0    |
>
> We also apply MixKV to a **long-context–tuned MoE LVLM**, Qwen3-VL-30B-A3B-Instruct in the revised manuscript, and observe consistent gains over SnapKV (**Table 7**), including a **+13.6** improvement on OCRBench at Budget = 64. These results together suggest that MixKV is robust, complementary to **architecture-level optimizations** (e.g., GQA, MoE), and compatible with **data-centric compression approaches** (e.g., token compression), remaining effective under realistic long-context LVLM settings.
>
> [1] An image is worth 1/2 tokens after layer 2: Plug-and-play inference acceleration for large vision-
> language models. ECCV, 2024.
>
> [2] Visionzip: Longer is better but not necessary in vision language model. CVPR, 2025.

---

> > ### Author Response · Authors · 2025-11-27
> > **Kind Reminder Regarding Rebuttal**
> >
> > Dear Reviewer eFa6,
> >
> > Thank you for your positive recognition of our work. We truly appreciate your supportive feedback and the time you’ve dedicated to reviewing our paper.
> >
> > As the rebuttal deadline approaches in about one week, we would like to kindly ask whether you have any further questions or points you’d like to discuss. If so, please don’t hesitate to let us know—we’re happy to provide additional clarification or details.
> >
> > Best regards,
> > The Authors

---

### Official Review · Reviewer_nwar · 2025-11-01

**Soundness:** 2
**Presentation:** 3
**Contribution:** 2
**Rating:** 4
**Confidence:** 5

**Summary:**

This paper proposes MixKV, a novel KV cache compression method that adaptively balances importance and diversity at the attention head level. MixKV quantifies semantic redundancy per head and mixes importance with diversity to select KV pairs for retention. Extensive experiments across multiple visual-text tasks and compression baselines show that MixKV consistently improves performance under tight memory budgets.

**Strengths:**

1. The paper provides a thorough analysis of semantic redundancy in LVLMs, highlighting a previously underexplored limitation of importance-only KV compression.
2. The adaptive mixing of importance and diversity per head is both intuitive and empirically validated.
3. MixKV can be integrated with existing compression methods without modifying the underlying compression operator, making it practical for real-world deployment.

**Weaknesses:**

1. The paper does not include a direct comparison with closely related methods such as [HeadKV](https://arxiv.org/abs/2410.19258),  which would strengthen the empirical evaluation.
2. The analysis of Cross-modality Redundancy Differences does not clearly illustrate the "cross-modality" aspect. From my understanding, it's more about *Two different modality*.
3. While the paper discusses Head-wise Redundancy Differences, similar redundancy patterns may also exist in pure LLMs. The unique characteristics or observations specific to VLMs should be clarified and emphasized.
4. In Section 4.2, it would be valuable to further discuss the observed patterns—specifically, why MixKV significantly improves baselines on Information Aggregation tasks but not on Information Localization tasks. Providing concrete examples would enhance the discussion.
5. The definition of "total latency" in Figure 5 is unclear. Details such as prompt length, number of decoding steps, and how compression overhead is measured should be explicitly disclosed.

**Questions:**

None

---

> ### Author Response · Authors · 2025-11-23
> **Response to Reviewer nwar (part 1)**
>
> Thank you for your recognition of our work and for raising several insightful questions. Your feedback has greatly helped us improve the quality of the paper. Below, we provide point-by-point responses to your comments.
>
> **W1. The paper does not include a direct comparison with closely related methods such as HeadKV, which would strengthen the empirical evaluation.**
>
> **A1.**   We thank the reviewer for the suggestion to compare with HeadKV. After carefully reading this work, we find that HeadKV is originally designed for LLMs processing **pure texts** and is conceptually similar to the LLM KV compression baselines we already compare against, such as SnapKV, AdaKV, and PyramidKV. In addition, HeadKV (similar to AdaKV and SparseMM) is a *head-wise budget allocation* method that reallocates KV cache budgets based on head importance. In contrast, **MixKV is not a budget allocation scheme**: it focuses on the redundancy heterogeneity across LVLM heads, assigning each head a redundancy score and using it to balance importance-based metrics (e.g., SnapKV, AdaKV, PyramidKV) with the diversity-based metric, so that semantically redundant heads can still retain complementary information.
>
>  As emphasized in the paper, MixKV is designed as a seamlessly pluggable module and already shows consistent gains when combined with SnapKV, AdaKV, and PyramidKV across multiple benchmarks and models (Tables 1/2/3), indicating that MixKV is largely orthogonal and complementary to head-wise budget allocation strategies such as HeadKV. Moreover, following the experimental setting of HeadKV (which only provides head-level importance score distributions on pure-text tasks), we include an **additional experiment with HeadKV+MixKV** on Mistral-7B-Instruct-v0.2 over LongBench and LooGLE in the revised paper (**Appendix A.4, Table 5**, marked in red). The experimental results are also shown in the table below and demonstrate that MixKV further improves HeadKV across long-context pure-text tasks; in particular, it yields consistent gains on all tasks under the constrained KV cache budget of 128. This further highlights the generalizability of our MixKV framework.
>
>
> | Methods             | Single-Doc QA       |               |               | Multi-Doc QA        |               |               | Avg.   | Long Dependency QA  |                   |           |               | Avg.  |
> |---------------------|---------------------|---------------|---------------|---------------------|---------------|---------------|--------|---------------------|-------------------|-----------|---------------|-------|
> |                     | NartvQA             | Qasper        | MF-en         | HotpotQA            | 2WikiMQA      | Musique       |        | Doc.QA              | Info. Retrieval   | Timeline  | Computation   |       |
> | Full KV             | 26.63               | 32.99         | 49.34         | 42.77               | 27.35         | 18.78         | 32.98  | 12.17               | 15.52             | 0.49      | 10.03         | 9.55  |
> | **Budget = 1024**   |                     |               |               |                     |               |               |        |                     |                   |           |               |       |
> | HeadKV              | 25.88               | **31.28**     | **50.54**         | 40.61               | 27.57         | 18.80         | 32.45  | 11.93               | **14.87**         | 0.49      | **9.56**      | **9.21** |
> | + MixKV             | **26.26**           | 31.20         | 50.07     | **40.99**           | **27.88**     | **19.93**     | **32.72** | **12.06**         | 14.73             | **0.50**  | 9.33          | 9.16  |
> | **Budget = 128**    |                     |               |               |                     |               |               |        |                     |                   |           |               |       |
> | HeadKV              | 24.34               | 26.60         | 48.55         | 40.69               | 25.97         | 15.34         | 30.25  | 10.48               | 12.72             | 0.53      | 10.04         | 8.44  |
> | + MixKV             | **24.39**           | **27.70**     | **49.85**     | **42.48**           | **27.21**     | **15.40**     | **31.17** | **10.62**         | **13.08**         | **0.73**  | **10.31**     | **8.69** |

---

> ### Author Response · Authors · 2025-11-23
> **Response to Reviewer nwar (part 2)**
>
> **W2. The analysis of Cross-modality Redundancy Differences does not clearly illustrate the "cross-modality" aspect. From my understanding, it's more about Two different modality.**
>
> **A2.** We appreciate the reviewer’s clarification. Your understanding is correct: our analysis focuses on how the KV cache exhibits different redundancy patterns when the model processes **vision** versus **language** inputs, and we observe stronger redundancy on the visual side. The original term “Cross-modality Redundancy Differences” may indeed suggest a broader cross-modal setting than what we actually study.
>
> To avoid confusion, we have revised the terminology in the paper to **“Vision-Language Redundancy Differences”** and updated the corresponding text (**Introduction**, marked in red) to explicitly state that our analysis concerns redundancy differences between the vision and language modalities in LVLMs. We thank the reviewer again for this helpful suggestion.
>
>
>
> **W3. While the paper discusses Head-wise Redundancy Differences, similar redundancy patterns may also exist in pure LLMs. The unique characteristics or observations specific to VLMs should be clarified and emphasized.**
>
> **A3.** We thank the reviewer for this insightful comment. We agree that head-wise redundancy patterns may also appear in pure LLMs. To better highlight VLM-specific characteristics, we have extended our analysis by adding **a pure LLM baseline** (Qwen2) processing only text, alongside its LVLM counterpart (Qwen2-VL) processing both pure-text and vision-language inputs, in the updated **Figure 5 in Appendix A.2**.
>
>  Concretely, we sample 100 examples from LongBench and 100 examples from TextVQA to compute head-wise redundancy and plot the updated Figure 5. From this visualization, we observe that heads at the same positions exhibit similar redundancy trends for Qwen2-VL and Qwen2, but the average redundancy of Qwen2-VL on vision-language data is consistently higher than that of both Qwen2 and Qwen2-VL on pure-text data. This indicates that while both text and vision-language inputs contain some redundancy, visual signals are inherently more redundant (e.g., due to repeated or highly similar visual patterns). This is consistent with Figure 1 and our first empirical finding (“Vision-Language Redundancy Differences”): vision-language modeling introduces stronger head-wise redundancy compared to text-only LLMs. We have added more detailed discussion of these observations in **Appendix A.2** (“More Discussions on Redundancy Differences”, highlighted in red in the revised manuscript).
>
> **W4. In Section 4.2, it would be valuable to further discuss the observed patterns—specifically, why MixKV significantly improves baselines on Information Aggregation tasks but not on Information Localization tasks. Providing concrete examples would enhance the discussion.**
>
> **A4.** We thank the reviewer for the suggestion. As briefly discussed in Section 4.2, MixKV is designed to jointly consider importance and diversity of retained KV entries, which improves global information coverage. Information Aggregation tasks require integrating evidence from multiple distant parts of the context, so better global coverage naturally brings larger gains.
>
>   In contrast, Information Localization tasks mainly depend on whether the local region containing the answer is preserved; once that span is kept, additional global diversity has limited benefit. Existing importance-based baselines already retain most high-importance tokens near the target span, so MixKV only yields modest improvements there. We will clarify this intuition in Section 4.2 and briefly connect it to our redundancy analysis (Figure 1), which also shows that MixKV provides particularly strong gains in more redundant vision-language settings, while still improving long-context text benchmarks.
>
>
> **A5. The definition of "total latency" in Figure 5 is unclear. Details such as prompt length, number of decoding steps, and how compression overhead is measured should be explicitly disclosed.**
>
> **A5.**   We thank the reviewer for pointing this out. Following SparseMM, we report efficiency at a fixed long-context setting. Concretely, in Figures 4 and 5 we set the **prompt length to 32K tokens** and fix the **output length to 100 decoding steps** when applying different KV cache budgets (64/128/256 for the corresponding curves). The reported **“total latency”** is the **end-to-end time** for the model to process the 32K-token input and generate 100 output tokens, where all KV cache compression operations are performed online and their overhead is fully included in this measured latency.

---

> > ### Author Response · Authors · 2025-11-27
> > **Kind Reminder Regarding Rebuttal**
> >
> > Dear Reviewer nwar,
> >
> > Thank you again for your thoughtful and high-quality feedback. Your comments have already helped us significantly improve the quality of our paper. We have responded to your concerns and revised the manuscript in line with your insightful suggestions.
> >
> > With approximately one week remaining until the rebuttal deadline, we have not yet received any follow-up questions. If you need further clarification or additional information, please do not hesitate to reach out. We are fully prepared to provide all necessary details.
> >
> > Best regards,
> > The Authors

---

### Author Response · Authors · 2025-11-29
**Summary of Rebuttal**

We would like to express our sincere gratitude to all the reviewers for their thorough evaluation and valuable feedback. During the rebuttal period, we focused on providing additional results that demonstrate the applicability, generalizability, and strong performance of our method. We also conducted further analyses to enrich the new findings and clarify key aspects of our method, inspired by the reviewers' insightful questions. Specifically, these include:

### **More Experimental Results:**

- **Extension to larger LVLMs and MoE-style LVLMs (for Reviewers vSGZ and eFa6):** We applied MixKV to larger LVLMs and MoE-style models, specifically InternVL3-38B and Qwen3-VL-30B-A3B-Instruct, alongside baseline methods. The results show that MixKV improves performance consistently across these models, demonstrating its robustness in advanced LVLMs.

- **Integration with HeadKV (for Reviewer nwar) and evaluation on long-context tasks (for Reviewers vSGZ and eFa6):** We integrated HeadKV within the MixKV framework and evaluated it on LongBench and LooGLE, covering 10 long-context sub-tasks. MixKV effectively enhances HeadKV's performance on long-text tasks, particularly under constrained KV budgets, demonstrating its robustness in long-context tasks as well.

- **Evaluation on video tasks (for Reviewer vSGZ and nMmG):** We extended MixKV to VATEX, a video captioning benchmark, to evaluate its performance on video tasks. The results show that MixKV continues to improve baseline methods, highlighting its effectiveness in image, video, and long-context pure-text tasks.

- **Comparison and combination with token compression methods (for Reviewer eFa6 and vSGZ):** We combined token compression methods (e.g., FastV, VisionZip) with MixKV and evaluated them on multimodal benchmarks. While token compression can reduce context length, it leads to performance degradation. In contrast, combining it with MixKV reduces both context length and KV cache memory while maintaining performance.

### **Further Discussions:**

- **More discussion on new findings (for Reviewer nwar and vSGZ):** We added Appendix A.2 to further discuss redundancy differences identified in the Introduction, including Vision-Language and Head-wise Redundancy Differences. Visualizing Qwen2 and Qwen2-VL processing pure-text and vision-language data shows that semantic redundancy is much weaker in pure-text data, supporting the validity of our new findings and the need for the "mixing importance with diversity" design in MixKV.

- **Broader discussion on other KV cache (or GPU memory) saving techniques (for Reviewer eFa6):** We expanded Section 2 with a broader discussion on long-context optimization and GPU memory-saving techniques, covering three main areas: (i) Efficient computational architectures, (ii) Model-centric compression, and (iii) Data-centric compression, analyzing how each contributes to model efficiency.

### **Detailed Clarifications:**

- **More detailed introduction of experimental settings (for Reviewer eFa6 and nMmG):** We have provided further details on the experimental setup, including specifics of the efficiency analysis and the concepts in the ablation study (e.g., "online head weights") to offer a deeper understanding of our work.

* **Clear distinction from previous works (for Reviewer vSGZ):** We have clarified the differences between our work and prior studies. MMinference identified sparse attention patterns across modalities and proposed a plug-and-play sparse attention method. VisionZip discovered visual token sparsity at the vision encoder output and proposed a visual token compression method. In contrast, MixKV identifies heterogeneous head-wise semantic redundancy within LVLMs and leverages this by analyzing redundancy across different heads. By combining importance and diversity within the KV cache, MixKV preserves more semantic information during compression. Unlike previous works that focus on attention patterns or token compression, our approach focuses on KV pair redundancy. This enables improved inference efficiency on both multimodal and pure-text tasks, and it integrates well with existing methods like FastV and VisionZip to further enhance their performance.


We believe that these additional results, detailed analyses, and clarifications have thoroughly addressed the concerns raised by the reviewers. We are grateful for their constructive suggestions, which have helped us significantly improve our paper.

---

### Meta-Review · Area_Chair_jHup · 2026-01-02

**Summary:**

This paper proposes MixKV, a plug-and-play KV cache compression framework for LVLMs that jointly considers importance and diversity, with adaptive head-wise mixing based on estimated semantic redundancy. Extensive experiments across multiple visual-text tasks and compression baselines show that MixKV consistently improves performance under tight memory budgets.

All reviewers acknowledge that the paper is supported by very extensive experimental results and that its motivation is well-defined. Although Reviewer nwar and Reviewer vSGZ initially assigned a score of 4, the authors provided detailed responses during the rebuttal phase. After carefully reviewing the rebuttal discussion, I believe that the reviewers’ concerns have been addressed. Therefore, I think this paper is above the acceptance threshold.

**Reviewer Concerns:**

Reviewer nwar and Reviewer vSGZ

**Reviewer Scores:**

The authors provided detailed responses for Reviewer nwar and Reviewer vSGZ's comments during the rebuttal phase.

---

### Decision · Program_Chairs · 2026-01-26

Accept (Poster)